# Breakage fusion bridge cycles drive high oncogene number with moderate intratumoural heterogeneity

Oncogene amplification is a key driver of cancer pathogenesis. Both breakage fusion bridge (BFB) cycles and extrachromosomal DNA (ecDNA) can lead to high oncogene copy numbers, but the impact of BFB amplifications on intratumoral heterogeneity, treatment response, and patient survival remains poorly understood due to detection challenges with DNA sequencing. We introduce an algorithm, OM2BFB, designed to detect and reconstruct BFB amplifications using optical genome mapping (OGM). OM2BFB demonstrates high precision (>93%) and recall (92%) in identifying BFB amplifications across cancer cell lines, patient-derived xenograft models, and primary tumors. Comparisons using OGM reveal that BFB detection with our AmpliconSuite toolkit for short-read sequencing also achieves high precision, though with reduced sensitivity. We identify 371 BFB events through whole genome sequencing of 2557 primary tumors and cancer cell lines. BFB amplifications are prevalent in cervical, head and neck, lung, and esophageal cancers, but rare in brain cancers. Genes amplified through BFB exhibit lower expression variance, with limited potential for regulatory adaptation compared to ecDNA-amplified genes. Tumors with BFB amplifications (BFB(+)) show reduced structural heterogeneity in amplicons and delayed resistance onset relative to ecDNA(+) tumors. These findings highlight ecDNA and BFB amplifications as distinct oncogene amplification mechanisms with differing biological characteristics, suggesting distinct avenues for therapeutic intervention.

Somatic copy number amplification of tumour-promoting oncogenes is a major driver of cancer pathogenesis[1]. High copy amplifications (CN > 8) are typically localized to specific genomic regions as focal amplifications (fCNA)[2,3]. Currently, two main mechanisms for high copy number oncogene amplification predominate: extrachromosomal DNA (ecDNA)[4,5] and breakage fusion bridge (BFB) cycles[6–8]. The independent replication of ecDNAs, their random segregation into daughter cells, and the positive selection for a higher (or appropriate) number of proliferative elements (e.g., oncogenes) provides the genetic basis for the rapid modulation of oncogene copy numbers in cells and explains much of the focal oncogene amplifications observed in cancer[4,9]. In addition, the circular shape of ecDNAs

alter their chromatin accessibility, remodels their physical structure, and interactions of the DNA and its topological domains[10–13]. The remodeled structure changes the epigenetic landscape, generates new long-range cis-regulatory interactions, and enhances oncogene transcription to drive tumour pathogenesis. The rapid modulation of DNA copy number mediates resistance to targeted therapy via multiple mechanisms. These include: an increase in the number of copies through selection for higher numbers of ecDNA per cell[14], a stabilization in copy number by reintegration of ecDNA into non-native chromosomal locations, or through the biogenesis of entirely new, compensating ecDNA[15]. Patients whose tumour genomes carry ecDNA are known to have worse outcomes relative to those with ecDNA(-)

✉ e-mail: Jean_Zhao@dfci.harvard.edu; pmischel@stanford.edu; vbafna@ucsd.edu

samples[5]. For these reasons, identification of ecDNA, and vulnerabilities specific to ecDNA(+) tumours remains an important challenge for understanding cancer biology.

The other main mechanism linked to high copy number oncogene amplification in cancer, Breakage-Fusion-Bridge (BFB) cycles, was proposed nearly 80 years ago by Barbara McClintock to explain patterns of genomic variation in irradiated maize cells[16,17]. BFB cycles start during a bridge formation (usually between sister chromatids) as a stabilizing repair intermediate for DNA breaks or telomere loss[18]. Unequal mitotic separation and breakage of the bridged chromosomes creates an inverted duplication on one chromosome, and a deletion on the other. The broken ends result in continued BFB cycles until the telomere is re-capped[19,20]. As with ecDNA, the lengthened chromosome may contain an oncogene that provides a proliferative advantage and selection for high copy number chromosomes creates rapid copy number amplification through successive BFB cycles. Other rearrangements might accompany BFB, confounding detection.

Recent experiments with human cell lines have shed light on the role of BFB in genome instability. Dicentric chromosomes generated by telomere dysfunction formed anaphase bridges that persisted intact through mitosis to form DNA bridges that were broken during interphase ultimately resulting in Kataegis and other catastrophic events such as chromothripsis[21]. Broken chromosome bridge fragments lag during anaphase and are frequently missegregated, forming micronuclei, and promoting additional chromothripsis[22]. In another interesting experiment where HeLa cell-lines had acquired resistance through stepwise increase of methotrexate, breakage fusion bridge cycles were observed, along with ecDNA formation and chromosome shattering/chromothripsis[23]. Together, these experiments suggested that initial BFB formation had stochastic outcomes that could include chromothripsis and ecDNA formation (Supplementary Fig. 1), but also stable BFB cycles that manifested as focal HSR amplifications on the native chromosome[23]. Detecting and characterizing stable BFB cycles is a goal of our paper.

BFB cycles are an important driver of oncogene amplification in cancer. In a pan cancer cytogenetic study, anaphase bridges and dicentric chromosomes were identified in 41 of 45 tumour samples but rarely in normal fibroblasts[24]. *HER2* positive breast cancers revealed a significant enrichment of BFB signatures and ecDNA within amplified *HER2* genomic segments[6,7]. Experimental work has also revealed mechanistic aspects of BFB cycles. Mice that were deficient in both non-homologous end joining (NHEJ) DNA repair protein(s) and *TP53* developed lymphomas that harbored BFB amplification of IgH/c-myc[25]. Bridge breakage was attributed to mechanical tension on structurally fragile sites, and the bridges were most frequently severed in their middle irrespective of their lengths[26]. Repeat-mediated genomic architecture surrounding the *ERBB2* (*HER2*) locus was implicated in promoting BFB cycles[6].

These results are indicative of a unique and important role for BFB cycles in cancer progression. However, the scope and extent of BFB amplification in cancer is not completely known, including the role of fragile regions in mediating breaks of the dicentric bridge and the consequent clustering of BFB structures. Moreover, it is not clear if the different modes of oncogene amplification including BFB cycles, ecDNA accumulation, and other intrachromosomal rearrangements are functionally interchangeable for cancer progression, if they amplify the same oncogenes, and whether they are prevalent in the same cancer subtypes. These questions require a systematic survey of BFB amplifications across many cancers, which in turn demands methods for reliable detection of BFB amplifications from genomic data.

In this work, we addressed several of the aforementioned questions. Specifically, we developed a method, OM2BFB, for the detection of BFB and the characterization of their structure using BioNano Optical Genome Maps (OGMs)[27]. The choice of the OGM technology was largely motivated by its relatively low cost, high coverage, and exceptionally high contig lengths (N50 length 38.4 Mbp). We validated the OGM based detection of BFB(+) amplification mechanisms through extensive cytogenetics experiments. Next, using OM2BFB as the standard, we benchmarked short-read based BFB detection, using our previously developed AmpliconSuite (AS) toolkit[28,29]. We applied OM2BFB and AS on 1538 whole genome samples to identify the location, scope, gene content and structural aspects of BFB based focal amplification. Finally, we integrated functional data including chromatin conformation, response to targeted drug treatment, gene expression, and patient outcomes to gain insight into how BFB cycles contribute to diverse cancer phenotypes.

## Results

A likely BFB-originating mechanism is the fusion of sister chromatids after a double strand break[18]. This can generate dicentric chromosomes that are subject to iterative cycles of breakage and fusion terminating with telomere restoration[6,19,30]. To formalize BFB cycle amplification, denote a chromosomal arm using consecutive genomic segments A, B, C, D, starting from the centromere and going towards the telomere. A double strand break removes segment D (Supplementary Fig. 2[18]). In a pure BFB cycle, where only a single chromosome is implicated, we could see a bridge formation, leading to the di-centric arm $ABC\underline{CBA}$, with the bar representing an inversion of the genomic segment. Subsequent breakage between B and A leads to a genome $ABC\underline{CB}$, which carries an inverted duplication, and a broken end, allowing for the process to repeat. A small number of BFB cycles lead to a highly rearranged genome. For example,

$$ABC \rightarrow ABC\underline{CB} \rightarrow ABC\underline{CB}B \rightarrow ABC\underline{CB}BB\underline{BCC}B$$

Sampling genomic sequences from a BFB-rearranged genome, and mapping them back to the human reference leads to a characteristic BFB signature of ladder-like copy number amplifications and an abundance of foldback structural variations. These signatures have been used to detect BFB using genomic data[6,7,30]. The signatures are, however, not definitive, and many non-BFB amplifications, including ecDNAs, can also carry them. For example, the signature $ABBBBBBCCCC$, where each letter denotes a genomic segment, and the bars indicate inversions, shows amplification and foldbacks, but is not explained by BFB cycles.

Here, we propose OM2BFB, a BFB detection method (Fig. 1A). OM2BFB utilizes the ultralong optical maps to better identify foldback reads and copy number profiles[31]. Similar to previous methods, OM2BFB locates focal amplifications with minimum copy number and an abundance of foldback reads as candidate regions (Methods). For each candidate region, it then enumerates multiple possible BFB architectures, modifying methods that were previously developed, including by us, for short read whole genome sequences[32–34]. The candidate BFB cycles induce a copy number profile and foldbacks. The distinctive feature of OM2BFB is an optimization to algorithmically generate a sequence of BFB cycles that minimize the discrepancy between the predicted and observed copy numbers and foldbacks (Methods). A low score implies that the observed copy number and foldbacks can be strongly attributed to the predicted BFB cycle. High likelihood (low scoring) reconstructions are output, along with the score, in a stylized format (Fig. 1B).

We tested the performance of OM2BFB using extensive simulations. To enable this, we developed a BFB simulation method and used it to simulate 595 BFB-positive data sets, spanning a large number of segments and copy numbers. Additionally, we used ecSimulator[35] to generate 1198 BFB-negative cases. The cases were generated in a multitude of contexts that included chromothripsis and ecDNA. The positive and negative examples largely overlapped in the parameter space (Supplementary Fig. 3). However, ecDNA based amplifications

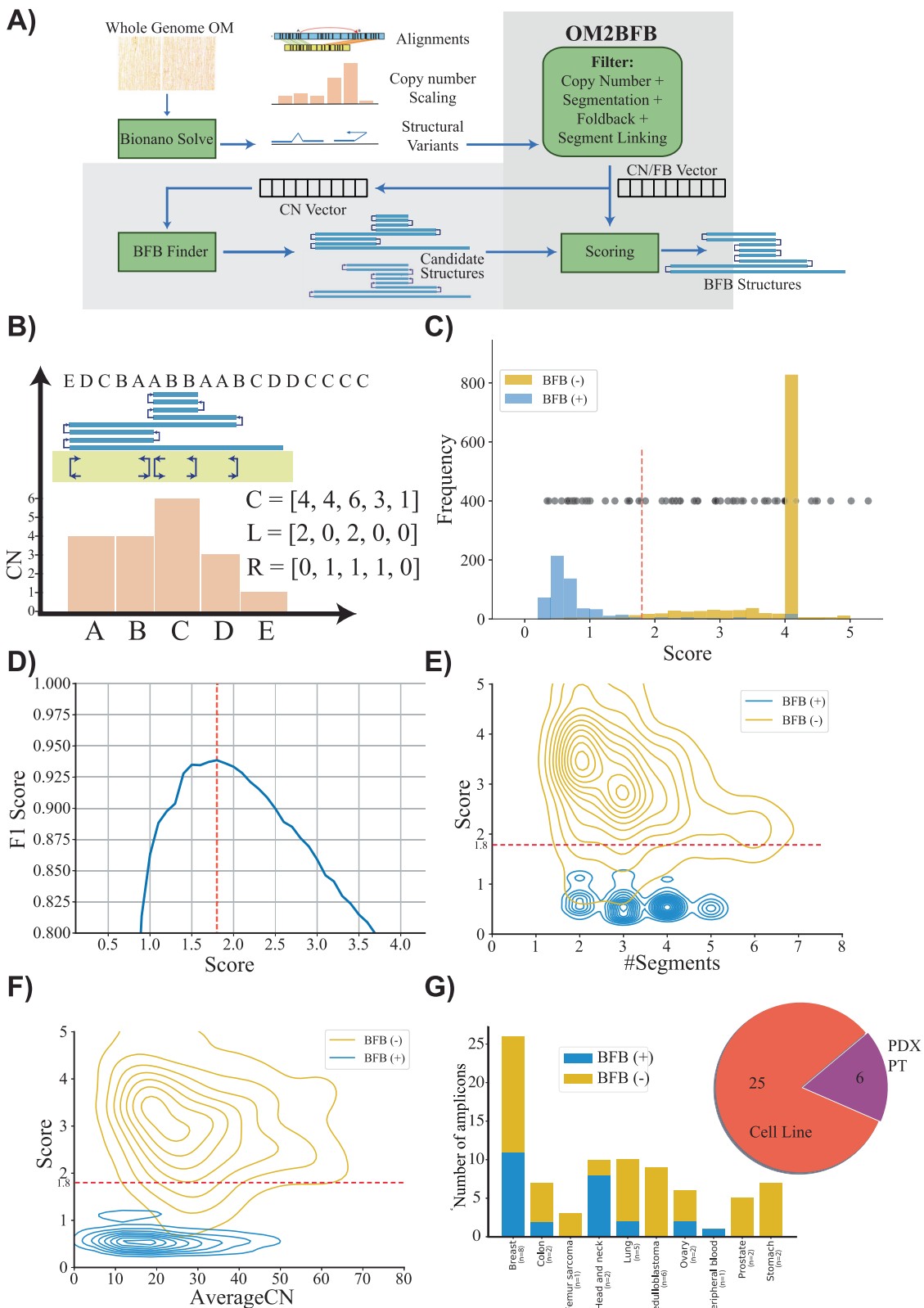

generally had higher copy numbers relative to BFB amplifications, consistent with current knowledge. OM2BFB scoring clearly separated the positive and negative examples (Fig. 1C). The maximum F1 score, describing the harmonic mean of the precision and recall, equaled 0.96 at a score-cutoff of 1.8 (Fig. 1D). Notably, 809 of the BFB negative examples either failed the initial filtering or represented a single segment with left and right foldbacks (deemed to be an ecDNA; Fig. 1C).

389 of the negative examples passed the initial filters. These represent focal amplifications with foldbacks and some would have been identified as BFB by available BFB calling tools using currently accepted metrics. However, OM2BFB correctly rejected 345 of the 389 negative examples at the score cutoff. We noted that the score distribution stayed consistent with changes in the number of segments from 1 to 7, and also with variation in the average copy number from 5 to 80

**Fig. 1 | Detecting and reconstructing BFB amplicons using Optical Genome Maps. A** Workflow of OM2BFB pipeline depicting the sequential steps involved in the analysis (Methods). **B** Schematic representation of OM2BFB output. The axes display genomic coordinates (x-axis) and copy-number (CN) (y-axis), with a separate track showing foldbacks. Orange bars represent segments with their respective copy numbers, navy arrows on the yellow background indicate foldback reads, and the blue rectangle on top represents the BFB structure described by an ordering of segment labels. The proposed BFB structure can be read by marking the segments traversed by the blue line, starting from the centromeric end. The vectors, C, L, and R refer to the copy number, left-, and right-foldback SVs that support the BFB. **C** Distribution of OM2BFB scores for 1198 simulated BFB negative (BFB(-)) cases and 595 simulated positive (BFB(+)) cases. Notably, 809 negative cases did not meet the filtering threshold and were reported with a high score (4) by OM2BFB. Black dots (placed at an arbitrary position on the y-axis for ease of visualization) represent the OM2BFB scores of 84 amplicons obtained from 31 cell lines. The red line at 1.8 marks the threshold

from (**D**) for defining BFB(+) from BFB(-) cases. Source data are provided as a Source Data file. **D** The F1 scores of OM2BFB, measured for different score cut-offs. The highest F1 score was achieved at a threshold of 1.8, and that score was selected as the threshold for separating BFB (+) from BFB(-) cases. Source data are provided as a Source Data file. **E** Distribution of OM2BFB scores across the number of segments in the simulated cases. The BFB(+) sample scores are independent of the number of segments, while BFB(-) samples reveal a slight bias of decreasing scores with higher number of segments. Source data are provided as a Source Data file. **F** Distribution of OM2BFB scores across the average segments' copy numbers in the simulated cases shows independence between the score and average copy number. Source data are provided as a Source Data file. **G** (Right) Distribution of OGM data types from 31 samples that include cancer cell lines, PDX models, and primary tumours (PT). (Left) Distribution of BFB(+) and BFB(-) cases across different cancer subtypes. Source data are provided as a Source Data file.

(Fig. 1E, F). Different components of the score, corresponding to segmentation, and fitting of the observed copy numbers and foldbacks to the candidate structures all contributed to the final performance (Supplementary Fig. 4). The small numbers of false negative examples could be attributed to missing foldback reads, while false positives were mainly due to ecDNA structures that strongly resembled BFBs in their shape (Supplementary Fig. 5).

We next applied OM2BFB to 84 amplicons using OGM data from 31 samples, including cancer cell lines, patient-derived xenograft (PDX) models, and primary tumours (Fig. 1G and Supplementary Data 1). Using the cut-off score of 1.8, 61 cases were identified as being BFB(-), and 23 as BFB(+), with high numbers of positive occurrences in HER2+ Breast cancer and Head and Neck cancer lines.

The presence of a homogeneously staining region (HSR) on a native chromosomal location is characteristic of BFB(+) structures, but other mechanisms might also generate native HSRs. Therefore, we used the criterion of "HSR amplification only on the native chromosome and an abundance of foldback reads," as the gold standard for validation[23]. Conversely, HSR amplification on a non-native chromosome with no amplification on the native chromosome is indicative of a BFB(-) structure, for example, through ecDNA formation and reintegration at a non-native locus[15]. We cytogenetically visualized 31 (21 metaphase, 10 interphase) structures using a probe from amplified regions (Supplementary Data 1). Applying the gold standard for metaphase cells, we obtained a validated set of 7 BFB and 14 non-BFB cycles. All 7 BFB cycles (Fig. 2A, B and Supplementary Figs. 6–12) and 13 of 14 non-BFB cycles were predicted accurately by OM2BFB (Fig. 2C, D and Supplementary Fig. 13). In each true-positive prediction, OM2BFB output a sequence of BFB cycles that completely supported the observed copy numbers and foldback reads, resulting in a low score (Greater than 1.8; Supplementary Figs. 6–12). We additionally validated the optical map predictions using whole genome nanopore sequencing for 4 BFB(+) samples. In each case, there was complete concordance between optical map and nanopore predicted copy numbers and foldbacks (Supplementary Figs. 6–9).

The BFB(+) structures included simple cases, such as a MYCL1 amplification in THP1 (Fig. 2A), but also complex ones, including a 10+Mbp BFB amplification in the HARA cell line that amplified PDHX1 on chr11p (Fig. 2B). The OGM data detected additional translocations from the short arm (p arm) to the long arm (q arm) of chromosome 11, and chr11q contained an amplification of CCND1. Using multi-FISH probes for PDHX1 (red), CCND1 (green), and chr11 centromere (yellow) (Fig. 2C), we confirmed that PDHX1 was amplified as a BFB on the native locus but also translocated to the q arm where it co-amplified CCND1.

OM2BFB gave a high score (>1.8) for 12 of 13 negative cases confirmed by metaphase FISH to be non-BFB. For example, it gave a score of 2.93 to the FGFR2 amplicon in SNU16 (Supplementary Fig. 13G). The amplicon is an ecDNA with multiple foldback reads. A more interesting

example was a MYC amplicon in the cell line H460. Despite extensive foldbacks, the OM2BFB score was 2.65 (Supplementary Fig. 13C). We performed a multi-FISH experiment probing for MYC (green), FGFR1 (red; near chr 8 centromere), and cen11 (yellow). Cytogenetics analysis clearly indicated two HSRs, one at the native locus on chr8, and the other on chr11 (Fig. 2C and Supplementary Fig. 13C). A careful reconstruction revealed a complex rearrangement of chr8 genomic segments with an insertion in chr11 (Fig. 2D). The amplicon contained multiple foldback SVs, but the patterns are unlikely to be the outcome of a BFB. More likely this represents a case of non-BFB related HSR.

## BFB structures show lack of heterogeneity in interphase cells from patient tumours

We tested OM2BFB in two separate patient data sets: the first dataset was a cohort of 6 samples, including primary tumour and patient-derived xenografts from individuals who had HER2(+) breast cancer with brain metastases (BCBM)[36,37]. The second data set consisted of untreated primary (HN137-Pri) and metastatic (HN137-Met) cell lines derived post-surgery from a patient with metastatic head and neck (HN) cancer[38]. Optical genome map data was acquired for these samples and analyzed using OM2BFB.

OM2BFB predicted 7 BFB amplicons in the BCBM data, including 2 containing HER2 (Supplementary Data 1 and Supplementary Fig. 14). In the HN137 data from one patient, OM2BFB identified 8 BFB amplicons. EGFR was amplified at a BFB site in both HN137Pri and HN137Met cell lines but increased in copy number from CN = 10–12 (Pri) to CN = 20 (Met) (Supplementary Fig. 15). Moreover, a BFB containing YAP1 and BIRC2 with high (CN = 40) amplification was present in HN137Met, but absent in HN137Pri. A BFB on chr11q, containing CCND1, was observed in both Pri and Met. A chr3q BFB containing EPHA1 was observed in HN137Pri, but not in Met. Finally, the copy number of a chr18p BFB was reduced from 30 to 10 and reduced in span.

As metaphase FISH was not possible for patient tumour cells, we used interphase analysis. We developed methods to segment the nuclei, identify and count FISH signals per nucleus. For BFB cycles and other HSRs, we would expect to see lower distinct foci counts and lower variability in the number of foci from cell to cell, in contrast to the higher counts and variance between cells for ecDNA(+) foci. We confirmed that the interphase foci had this property for the known BFB in HCC827 (EGFR), HSR in Colo320HSR (MYC), ecDNAs in Colo320DM (MYC) and MSTO211H (MYC) (Fig. 2E, F). For example, the ecDNA amplifications had variance in the number of foci exceeding 13.6, while all HSR and the BCBM cell lines had variance below 1.7. Thus, OM2BFB predictions were consistent with interphase FISH in 7 of 8 BCBM BFBs, with one false negative prediction, and in 2 of 2 HN137 cell lines with EGFR amplifications (Fig. 2F, Supplementary Fig. 14 and Supplementary Data 1). Intriguingly, DNA FISH probes for EGFR showed two stable foci per cell in HN137Met in contrast to a single HSR in HN137Pri

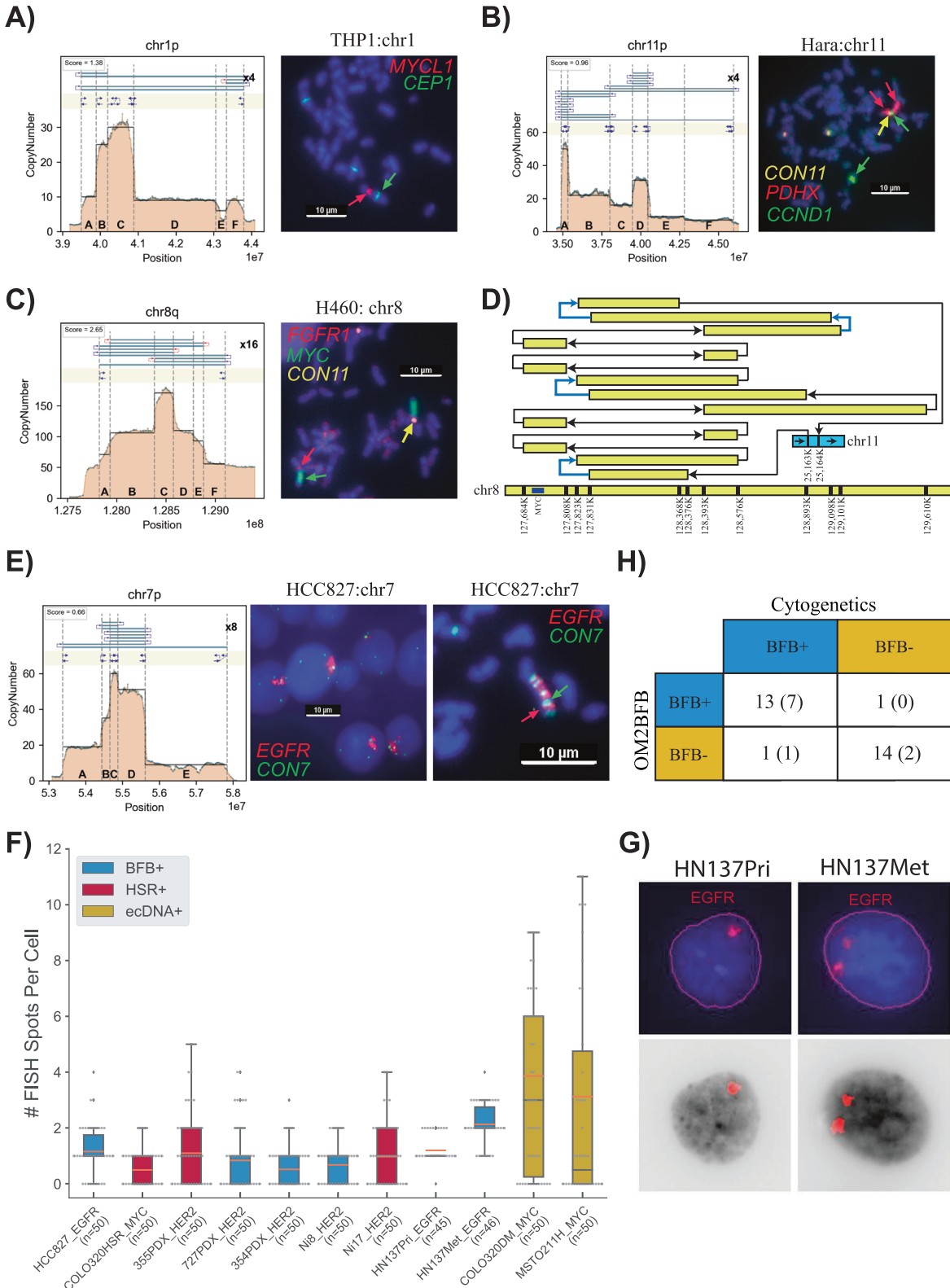

(Fig. 2F, G), explaining the doubled copy number in the metastatic line. The homogeneity of BFB structure between HN137-Pri and HN137-Met lines (Supplementary Fig. 15) suggested a stable transformation from primary to metastatic cells.

Taken together, the cytogenetics revealed that the presence of focal-amplification and foldback reads are not, by themselves, sufficient to predict BFB. They confirmed the utility of OM2BFB as a sensitive (13/14 = 92.8%) and precise (13/14 = 92.8%) method for BFB

prediction (Fig. 2H). Under a stricter condition, of observing HSR only on the native chromosome arm, we would have to treat HARA as a false positive and we would get sensitivity (12/13 = 92.3%) and precision (12/14 = 85.7%). Thus our results suggest that BFB driven amplicons are followed by additional rearrangements, including translocation to other loci and ecDNA formation. This is consistent with earlier results that suggest that anaphase bridge formation and dicentric breakage can be a precursor to gross DNA instability[22], including ecDNA

**Fig. 2 | Cytogenetic evidence for BFB and validation. A** Metaphase FISH image for the cancer cell line THP1, showcasing amplification on a native chromosome (chr1) with a score below 1.8 ($N = 11$). CEN1 is a probe for the centromere on chr1. **B** Metaphase FISH image for the cancer cell line HARA, displaying amplification on a native chromosome arm (chr11p) and also co-occurrence of HARA and *CCND1* on the q-arm, suggesting a duplicated translocation of the BFB site from the p arm to the q arm ($N = 24$). **C** Metaphase FISH image for the cancer cell line H460, demonstrating amplification on both native (chr 8) and non-native (chr 11) chromosomes with scores exceeding 1.8 ($N = 32$). **D** Visualization of HSR (homogeneously staining region) amplification on chr 8 (*MYC*) integrated in chr 11 in the H460 cell line. Blue arrows represent foldback reads within the structure. **E** Metaphase and Interphase FISH images for the cancer cell line HCC827, exhibiting amplification on a native chromosome with a score below 1.8. **F** Distribution of FISH Foci Count among cases with Interphase FISH images, highlighting lower number of foci and also lower variance in the number of foci in BFB and HSR cases compared to ecDNA cases. Center lines indicate the median, boxes represent the interquartile range (IQR) from the 25th to the 75th percentile, whiskers extend to the minimum and maximum values within 1.5 times the IQR. Source data are provided as a Source Data file. **G** Visualization of *EGFR* foci in interphase cells from HN137Pri and HN137Met lines. The top panel shows the original FISH image, while the bottom panel shows the computationally detected foci. **H** Summary of cytogenetic validation of OM2BFB calls. The number in parentheses refers to the number of interphase samples. This table includes $n = 22$ samples with available FISH data.

formation[23]. At the same time, there are many cases where BFB cycles result in stable focal amplification on the native chromosome, with low cell to cell heterogeneity.

## BFB amplifications are ubiquitous in multiple cancer subtypes

The paired end WGS-based Amplicon Suite (AS) pipeline, comprising Amplicon Architect (AA) and Classifier (AC), has been validated for ecDNA detection, but not for BFB detection, due to a lack of positive and negative examples. We compared the AC results on the same 83 amplicons initially scored by OM2BFB (Fig. 1C). Of the 23 BFB predictions made by OM2BFB, AC predicted 14 (Supplementary Data 1). Importantly, however, AC did not make any BFB(+) calls in the 61 amplicons that OM2BFB also labeled as BFB(-) (Supplementary Data 1). AC failed to predict some BFB cycles mainly due to a lack of foldback SV read identification using short-read data (Supplementary Fig. 16).

Next, we tested AC performance on 6 cytogenetically validated BFB and 12 non-BFB cycles (Supplementary Data 2). AC predicted 11 of 12 non-BFB structures as BFB(-) and 3 of 6 BFB cycles as BFB(+). Finally, we tested AC on another cohort of 9 samples where anaphase bridge structures had been experimentally observed[23]. Three of the 5 cases with stable BFB or BFB with some shattering were also predicted as BFB by AC (Supplementary Data 3 and Supplementary Fig. 17). Similarly, 4 cases shown to contain DMs (ecDNA) were also predicted as ecDNA(+) by AC. One case with only two BFB cycles did not pass the filter for focal amplification (Supplementary Fig. 18, and one case with BFB cycles followed by chromothripsis was predicted as BFB(-).

Altogether, these three different experiments suggest that the AC predictions of BFB have high precision (few false positives), and we utilized it to understand the landscape of BFB structures in large cancer genome repositories. We executed Amplicon Architect (AA) followed by AmpliconClassifier (AC) on 1,538 genomes from The Cancer Genome Atlas (TCGA), on 305 genomes consisting of premalignant Barrett's esophagus and esophageal cancer samples (BE)[39], and on 329 samples from the Cancer Cell Line Encyclopedia (CCLE)[40] (Fig. 3A). Despite the relatively lower sensitivity of AC to detect BFB in comparison to OM2BFB, the results still provided a significant overview of where BFB cycles occurred in the genome, and the genome structures of the BFBs. AC identified 371 BFB amplicons, including 258 in primary tumours, located across nearly every chromosome (Fig. 3B). Notably, the incidence in TCGA was markedly lower than in the BE and CCLE data. This can be attributed to the higher incidence of BFB in esophageal and lung cancers, and the high proportion of lung samples in CCLE data (Supplementary Data 4).

## BFB locations are highly dispersed, but not random

BFB structures were identified in multiple chromosomes, and did not show any preference for centromeric or telomeric locations (Fig. 3B, Supplementary Fig. 19, (two-tailed Binomial test, *p*-value = 0.48, test statistic = 68). We partitioned the genome into 5 Mb bins and marked the ones that carried BFB amplifications. Intriguingly, the BFB locations were not distributed randomly in those bins (Fig. 3C; two-tailed KS test, *p*-value = 3.7e-19, test statistic = 0.17). For example, a bin

comprising region K on chromosome 11 containing the genes *CTTN* and *CCND1* carried 27 of 371 BFB amplicons. Furthermore, while chromosomes 5 and 13 had fewer than 7 BFB events, chromosomes 3, 7, and 11 had more than 30 BFB events each. Multiple hypotheses explain the non-random distribution. The initiating break could occur at a fragile region of the genome. A recent result points to fragility in the *KRTAP1* region, 1.12 Mb telomeric to *HER2* as a cause for recurrent BFB amplification of *HER2*[41]. A second hypothesis is that the initial break is random, but subsequent amplification of an oncogene and positive selection for higher copy numbers leads to recurrent BFBs in specific genomic regions. To test these two hypotheses, we chose 7 genomic regions that were recurrently amplified via BFB at least 7 times. For each such region, we looked at 10 1 Mb windows telomeric to an oncogene to test if the breaks were preferentially clustered. In all cases but one, the breaks were randomly distributed, connected only by the fact of sharing a common amplified oncogene (Supplementary Fig. 20, Supplementary Data 5). The window containing the *KRTAP1* gene was indeed preferentially selected for *HER2* + BFBs (Fig. 3D; 16/30 breaks, nominal *p*-value 0.03; see methods permutation test). The *CCND1-CTTN* gene cluster on chromosome 11 also showed preferential break in a window immediately downstream, but did not reach the *p*-value = 0.05 level of significance. In the other 5 cases, there was no preference for any window in the 10 Mb region. While our data is sparse, the results suggest that an initial random breakage followed by positive selection for oncogene amplification can lead to a nonrandom distribution of BFB locations. Similarly, while the bridge breaks due to mechanical tension, a break immediately downstream of the oncogene is preferentially selected.

We observed that the distance between aligned ends of the foldbacks reads has a long tail, with over 65% of foldback distances greater than 1000 bp (Fig. 3E). The result suggests that fusion-bridge formation requires proximity, but not palindromic sequence.

## BFBs amplify the same oncogenes as ecDNA but are structurally different

We investigated the oncogenes amplified by BFB across all sample types. Consistent with a 'random breakage with selection' hypothesis, the BFB cycles amplified a large number of oncogenes. Intriguingly, the oncogenes amplified via BFB strongly overlapped with oncogenes amplified by ecDNA. Of the 52 oncogenes amplified at least 4 times on BFB cycles, 49 were also amplified as ecDNA at least once (Fig. 3F; Supplementary Data 6). In contrast, 19 (20%) of the 96 genes that were amplified at least 4 times as ecDNA were not found to be BFB amplified (*p*-value 0.03, Fisher Exact Test). These included *CDK4*, *MDM4*, *FGFR3*, and others. While we do not have a mechanistic hypothesis to explain this, we note that ecDNA can be formed through BFB-independent events, including episome formation and chromothripsis[9,22,23].

Positive selection for ecDNA can occur if they carry regulatory elements that serve as a 'roving enhancers' for genes on other ecDNA or chromosomes (*trans*)[42,43]. In contrast, the chromosome bound BFB structures, with locally mediated rearrangements, are unlikely to hijack distal enhancers. We hypothesized that selection for BFB structures

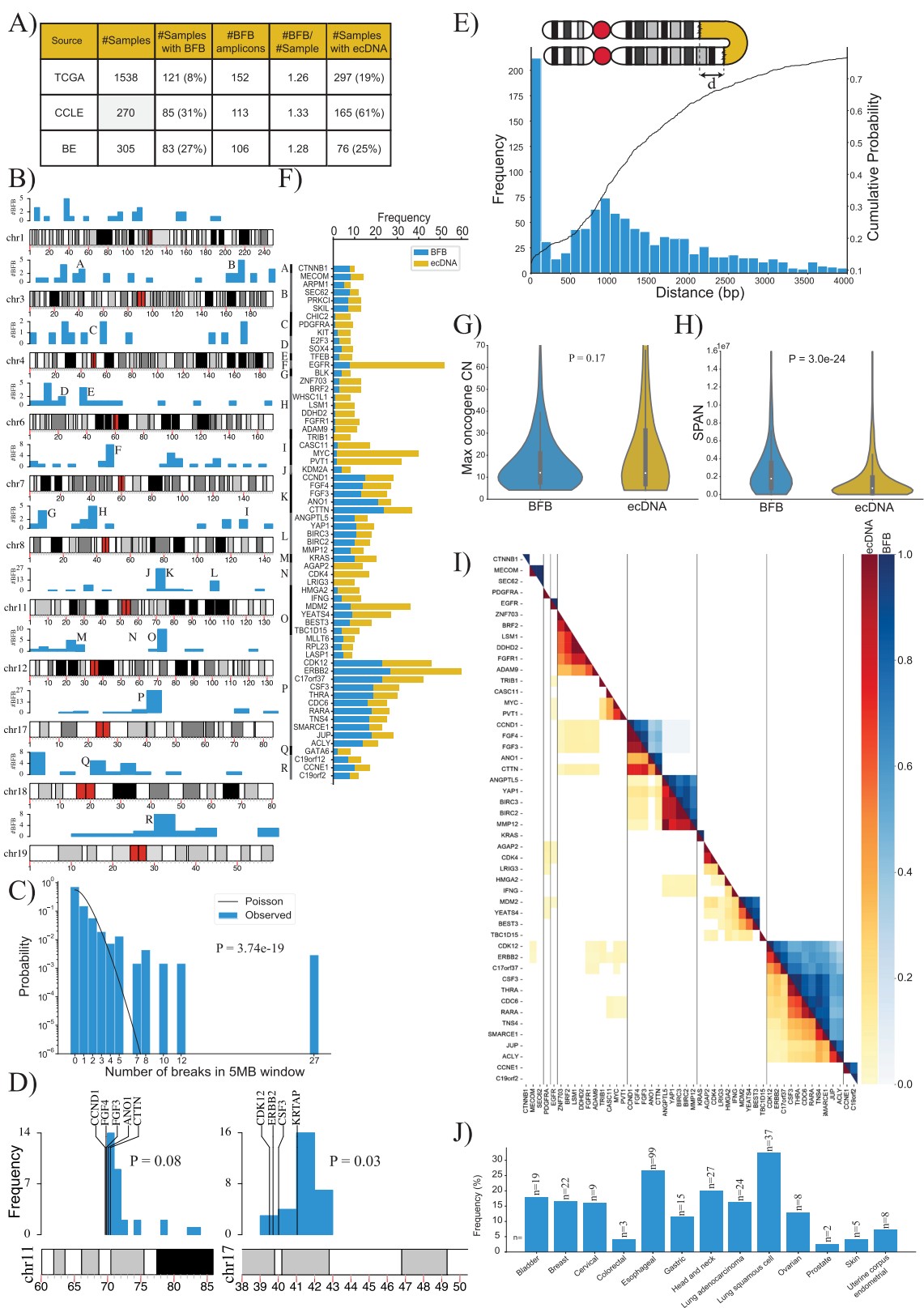

was largely mediated by amplification of oncogenes or *cis*-regulatory elements. Consistent with this hypothesis, we observed that only 5% (21 of 371) BFB amplicons did not carry a known oncogene (Supplementary Data 4) while over 10% (80 of 759) of ecDNA amplicons were oncogene free (*p*-value 0.003; Fisher exact test). The maximum oncogene copy numbers were, however, similar for BFB and ecDNA structures (Fig. 3G).

While BFB and ecDNA amplified the same genes, the amplicon structures were different. BFB amplifications had a larger span compared to ecDNA (Fig. 3H; mean span 2.8Mbp versus 1.5Mbp; two-tailed Rank-Sum test, *p*-value = 3.00e-24, test statistic = −10.09). Surprisingly, despite BFB structures having a large span, they did not co-amplify more proximal oncogenes than ecDNA. In fact, ecDNA structures were often multi-chromosomal, and co-amplified distant oncogenes (Fig. 3I

**Fig. 3 | The landscape of BFB amplifications in tumor samples. A** Summary of the number of BFB(+) samples among 2113 whole genome samples tested for BFB amplification using AmpliconSuite. The data was collected from three data sets: TCGA, BE[39] and CCLE[40]. **B** Locations of the first (most telomeric) break of the 371 BFBs in the human genome (hg38). Chromosomes with fewer than 12 BFBs are not shown. Source data are provided as a Source Data file. **C** The distribution of BFB occurrences (most telomeric break) in 5 Mbp windows compared against the Poisson distribution to test for randomness (two-tailed KS test, $p$-value = 3.7e-19, test statistic = 0.17). Source data are provided as a Source Data file. **D** The randomness of the first break in a 10 Mb region, telomeric to an amplified oncogene. Left panel: 29 BFBs on chr11 containing *CCND1*; Right panel: 30 BFBs on chr17 (ERBB2). *P*-values from BFB distributions calculated with one-tailed permutation-like test. Source data are provided as a Source Data file. **E** Distribution and cumulative distribution of the distance (d) between foldback reads. Source data are provided as a Source Data file. **F** Frequencies of the mode of amplification (BFB versus ecDNA) in oncogenes that are amplified at least 8 times in all datasets combined. Source data are provided as a Source Data file. **G** Violin plot showing the distributions of the maximum oncogene copy number between BFB ($n$ = 350) and ecDNA ($n$ = 677) amplicons (two-tailed Rank-Sum test, $p$-value = 0.176, test statistic = −0.93). Box plots within each violin indicate the median (center line), interquartile range (IQR) from the 25th to the 75th percentile (bounds of the box), and whiskers extending to the minimum and maximum values within 1.5 times the IQR. Source data are provided as a Source Data file. **H** Violin plot showing the distributions of amplicon length (SPAN) between BFB ($n$ = 391) and ecDNA ($n$ = 900) amplicons (two-tailed Rank-Sum test, $p$-value = 3.00e-24, test statistic = −10.09). Box plots within each violin indicate the median (center line), interquartile range (IQR) from the 25th to the 75th percentile (bounds of the box), and whiskers extending to the minimum and maximum values within 1.5 times the IQR. Source data are provided as a Source Data file. **I** Co-occurrence patterns of amplified oncogenes. Color-coded entry for (i, j) measures the fraction of times genes (i, j) were both amplified when either gene was amplified. The lower triangle shows ecDNA co-occurrence patterns and the upper triangle shows BFB co-occurrence patterns. Source data are provided as a Source Data file. **J** Distribution of BFB amplicons over different cancer subtypes. BFB amplicons were not found in brain and CNS related cancers, but were most abundant in lung and head and neck cancers. Source data are provided as a Source Data file.

lower triangle). Such co-amplification was rare in BFB (Fig. 3I upper triangle), because it would require the presence of independent BFB amplifications, or the formation of translocation bridges[44].

Similar to ecDNA, BFB amplicons were found in multiple cancer subtypes (Fig. 3J). Certain subtypes were more likely to carry BFB amplifications, including lung, esophageal, and head and neck cancers. However, BFBs were rare in brain cancers, where ecDNA amplifications are prevalent[5,13].

## BFB amplified genes show lower variance of gene expression, with fewer options for regulatory rewiring

We hypothesized that intrachromosomal BFB cycles were likely to function differently from the more mobile ecDNA (and even the ecDNA re-integrated as HSRs). Circularization and other rearrangements have been shown to change the topologically accessible domain (TAD) structure for ecDNA, changing their regulatory wiring, including the hijacking of distal enhancers[11–13]. In contrast, foldbacks are the primary source of rearrangements in BFB cycles. While foldbacks also change the conformation, they would be less likely to bring distal regions in close proximity. To test this hypothesis, we used HiChIP to measure chromatin interactions involving the enhancer- and promoter-associated mark H3K27ac on focal amplifications in the cell-line COLO320DM. The cell line amplified *MYC* (chr8) on ecDNA[4]. It carries another focal amplification on chr1, that originated with a duplication inversion characteristic of BFB[45]. Consistent with genomic rearrangements, the HiChIP interaction maps on both ecDNA and BFB regions showed a remarkable change in topological structure, when compared to the matching genomic regions in the control GM12878 line (Fig. 4A, B). For example, the rightmost foldback (Fig. 4B, green arrow) in the Chr1 amplicon topologically separates the telomeric region from the BFB region in COLO320DM, but not in GM12878. Similarly, the extensive rearrangements in the chr8 ecDNA create distant interactions in COLO320DM that are absent in GM12878. We next identified significant chromatin interactions in the two structures using NeoLoopFinder[46]. The 'neo-loops', marked by black spots, are interactions mediated by genomic rearrangements in the cell line, while the blue spots ('loops') are interactions attributable to conformational change (Fig. 4A, B). In both amplicons, we identified a larger number of distal (off-diagonal) interactions relative to GM12878. The number of distal H3K27ac-region interactions in the BFB amplification was larger relative to GM12878, but significantly lower than for the ecDNA amplification (Fig. 4C; two-tailed Peacock (2d-KS) test, $p$-value = 5.95e-05, test statistic = 0.728). To replicate these observations, we identified another cell line, H2170, which was cytogenetically validated to contain both a BFB and an ecDNA, and performed a Hi-C experiment with identical results (Supplementary Fig. 22). Thus, BFB driven

amplifications (unlike with ecDNA) could have fewer options for rewiring of the regulatory circuitry.

We also explored the normalized gene expression data from the cancer genome atlas for genes amplified on BFB and ecDNA. The expression of BFB amplified genes increased with copy number, similar to ecDNA (Supplementary Fig. 22), but there were important differences in transcription of other genes, especially relating to the immune response. EcDNA amplification has previously been associated with lower immune activity[5,47]. Using previously estimated cell type composition based on transcript evidence[47], we found that BFB(+) samples had increased concentration of multiple immune cell-types, including cytotoxic T cell (CD8+) levels and pro-inflammatory macrophages, relative to ecDNA(+) samples (Fig. 4D). Our results suggest that cancers with BFB amplification have lower immunosuppression than ecDNA amplifications, and might be more susceptible to checkpoint target engagement.

EcDNA(+) cells can rapidly modulate their copy number in response to changes in environment. For example, in a glioblastoma model, ecDNA integrated into chromosomes with reduced copy number upon drug treatment, and rapidly reappeared upon drug removal[43]. To test the mechanism of targeted therapy resistance in BFB-mediated amplifications, we obtained naive, drug resistant (Erlotinib and Lapatinib), and drug removed versions of the cell line HCC827, which amplifies *EGFR* on a BFB[28]. Metaphase FISH confirmed the stable BFB amplicon in naive, drug resistant, and drug removed lines (Fig. 4E). Remarkably, the *EGFR* copy number in Eb (Erlotinib) resistant lines was 30% lower than the naive and Erlotinib removed (ERDR) lines. These results were supported by DNA FISH experiments, which showed that the BFB was present in resistant cells, but only in 10 of 29 cells. In contrast, the amplification was universally observed in the untreated and drug removed lines. No changes were observed in the BFB amplification in Lapatinib drug resistant (LR) and Lapatinib removed (LRDR) cell lines (Supplementary Fig. 23) The results suggest that unlike ecDNA, BFB structures are more stable and sensitive to targeted therapy[48]. Drug resistance is likely mediated by a change in the population of cells carrying the BFB amplification rather than a change in its structure.

## BFB focal amplifications are more genomically stable than ecDNA focal amplifications

We observed significantly longer overall survival in patients with BFB (+)/ecDNA (-) relative to the ecDNA (+) cohort, out to 1100 days (Fig. 5A; two-tailed Log-rank test, $p$-value = 0.02, test statistic = 5.05). Although this difference was no longer seen at longer time points (Supplementary Fig. 24), these data raise the possibility that ecDNA may be linked to treatment resistance, which usually occurs earlier in

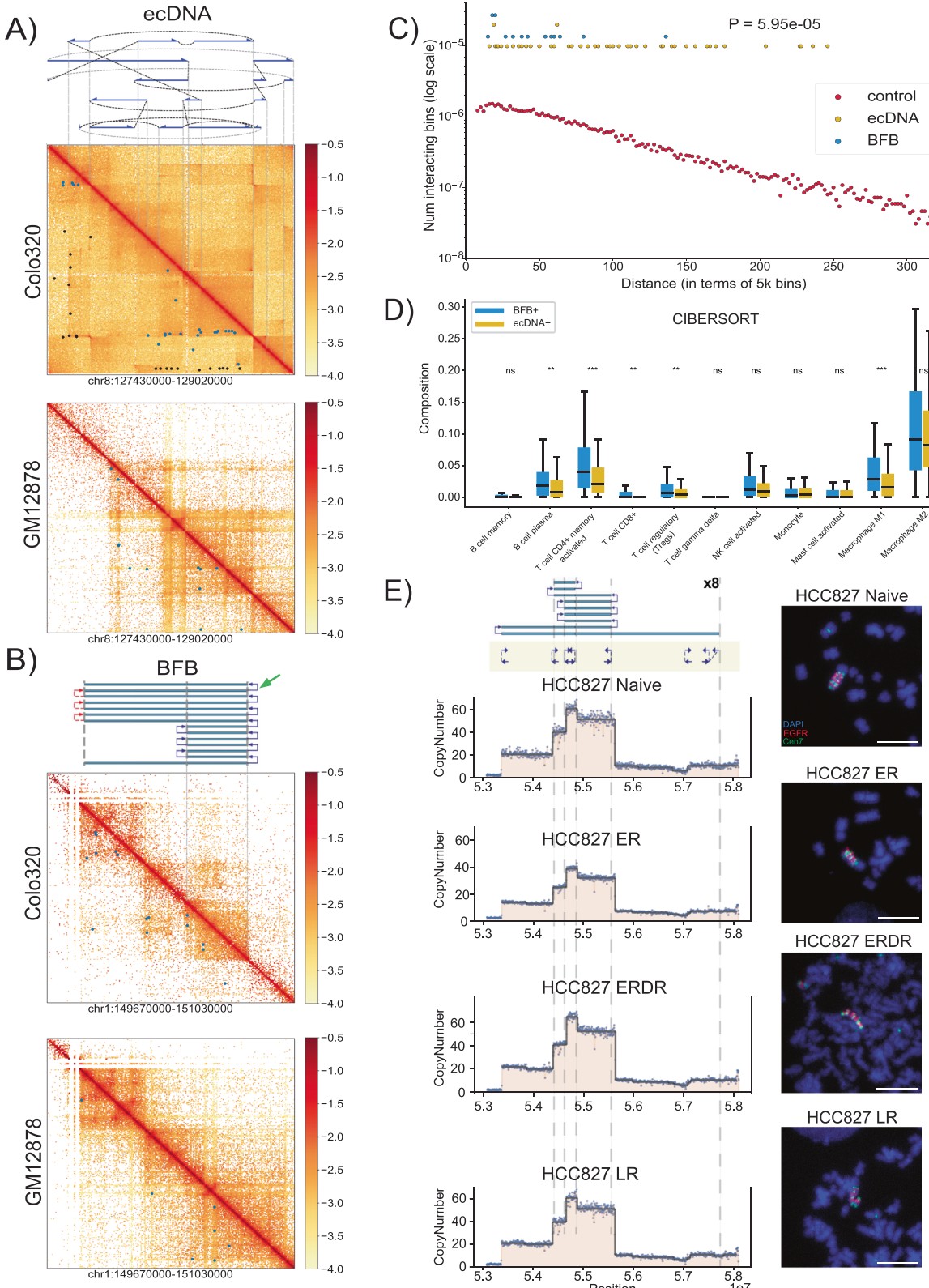

the course of the disease. Next, we asked if the BFB structures acquired additional rearrangements (became more complex) as the cancer progressed. In an earlier study, we developed a measure of amplicon complexity based on entropy of the number of genome segments in the focal amplification and the distribution of copy numbers assigned to genome paths extracted by AmpliconArchitect[39]. The manuscript also showed that patients with esophageal cancer (EAC) and patients

with the premalignant Barrett's esophagus (non-EAC) both carried ecDNA, but the complexity score was higher in the patients with cancer[39]. Similarly, we also identified BFB amplifications in both non-EAC and EAC patients. Surprisingly, the amplicon complexity in EAC BFBs was lower compared to the BFBs in premalignant non-EAC BFBs (Fig. 5B; *P*-value 0.001, two-tailed Mann-Whitney U test, statistics = 1345.0). Together, these results are consistent with BFB amplifications

**Fig. 4 | Structural and functional properties of BFB amplifications. A** Top: The structure of the *MYC*-amplified ecDNA from the COLO320DM cell line. Blue arrows indicate genomic segments from chr 8 amplified on the ecDNA; black dashed lines indicate SV breakpoints directly connecting two remote genomic segments; gray dashed lines indicate templated insertions of segments involving other chromosomes. Middle: Normalized HiChIP contact map of COLO320DM at the ecDNA locus. Colors indicate normalized contact frequencies from the most intensive (red) to the least intensive (yellow). Blue spots indicate significant chromatin interactions identified by NeoLoopFinder; while black spots indicate "neoloops" proximal to SV breakpoints and likely to be formed due to the genomic segments coming together in the cell line. Bottom: Normalized HiChIP contact map of GM12878 at the identical chr8 locus. Blue spots indicate significant chromatin interactions. Source data are provided as a Source Data file. **B** Top: The inferred structure of the BFB-like focal amplification from the COLO320DM cell line. Middle: Normalized HiChIP contact map of COLO320DM at the BFB locus. Colors indicate normalized contact frequencies from the most intensive (red) to the least intensive (yellow). Blue spots indicate significant chromatin interactions identified by Neo-LoopFinder. Bottom: Normalized HiChIP contact map of GM12878 at the identical chr1 locus. Blue spots indicate significant chromatin interactions. Source data are provided as a Source Data file. **C** Distribution of HiChIP interaction frequencies in

ecDNA and BFB-driven amplifications. For a specific genomic distance d (x-axis), the dot represents the fraction, among all pairs of genomic windows separated by d, of pairs with significant HiChIP interactions (2D Two-Sample Kolmogorov-Smirnov Test, *p*-value = 5.95e-05, test statistic = 0.728). Source data are provided as a Source Data file. **D** Differences in immune cell subtype compositions in BFB(+) cancers (*n* = 76) versus ecDNA(+) cancers (*n* = 297). (*$p < 0.05$; **$p < 0.01$; ***$p < 0.001$). Center lines indicate the median, boxes represent the interquartile range (IQR) from the 25th to the 75th percentile, whiskers extend to the minimum and maximum values within 1.5 times the IQR. *P*-values were calculated using a two-tailed rank-sum test. Exact *p*-values and statistics are provided in Supplementary Data 8. Source data are provided as a Source Data file. **E** Targeted Therapy Resistance of the HCC827 Cell Line continuing EGFR amplified within a BFB event. The top panel shows the BFB architecture in the HCC827 naive cell line along with metaphase FISH images (*N* = 17). Resistance formation to Erlotinib (ER) maintains the BFB amplicon structure, but the bulk copy number is highly reduced (*N* = 30). The copy number and the proportion of cells carrying the BFB signal are restored after drug removal (ERDR) (*N* = 6). No changes were observed in the BFB amplification in Lapatinib drug resistant (LR) line (*N* = 15). Scale bars in the fluorescent images represent 10 micrometers (μm).

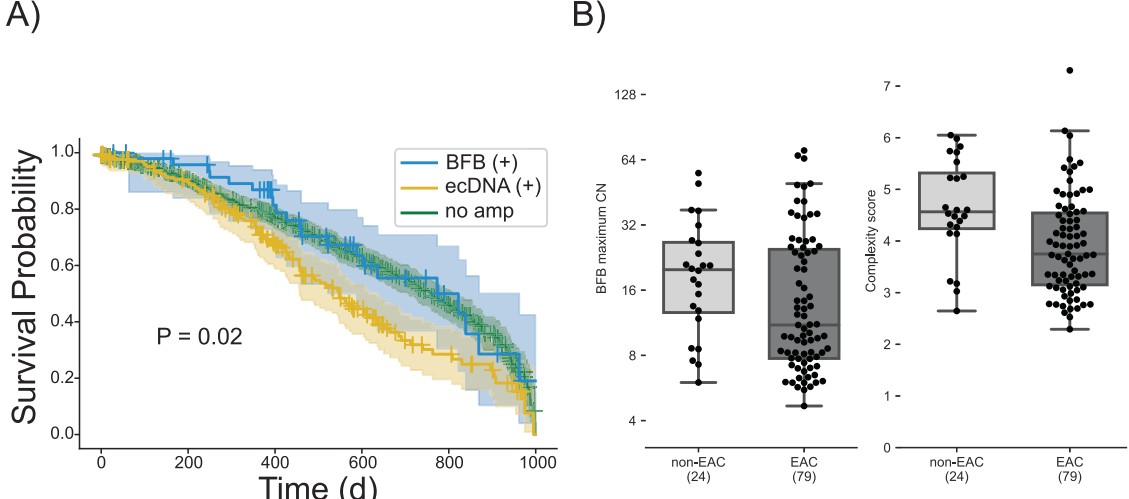

**Fig. 5 | Genome instability in BFB amplified genomes. A** Survival outcomes in the first 1000 days (d) for patients with BFB(+) but no ecDNA amplifications (*n* = 50) in their tumours compared to outcomes for ecDNA(+) patients (*n* = 171) show significant difference (two-tailed Log-rank test, *p*-value = 0.02, test statistic = 5.05). For comparisons, the survival outcome for patients (*n* = 600) with no amplification is also plotted. Error bands represent 95% confidence intervals for the survival probabilities. Patients with survival times exceeding 1000 days were excluded from

this analysis. Source data are provided as a Source Data file. **B** Maximum copy number and amplicon complexity scores for BFB amplicons sampled from Barrett's esophagus (non-EAC) compared to esophageal adenocarcinoma (EAC) patients. Center lines indicate the median, boxes represent the interquartile range (IQR) from the 25th to the 75th percentile, whiskers extend to the minimum and maximum values within 1.5 times the IQR. Source data are provided as a Source Data file.

being more stable and less adaptive than ecDNA amplification. As resistance is acquired through changes in allele frequency in populations of tumor cells, when compared to ecDNA(+) tumors, the decreased complexity and heterogeneity of BFB amplifications may lead to comparatively lower levels of intratumoral heterogeneity and slower resistance than their ecDNA(+) counterparts.

## Discussion

Initial cycles of bridge formation and breakage have varied outcomes. They can lead to genome instability, chromothripsis, and ecDNA formation. However, we specifically focus on the case where multiple BFB cycles on the native chromosome are followed by telomere reacquisition, without much additional structural variation. BFB is an important mechanism of focal oncogene amplification and has previously been observed in many cancers. However, accurate detection of BFB events and characterization of their architecture remains challenging[18]. We found that the typical signature of BFB events–a

ladder-like amplification pattern and an excess of foldbacks–may not be sufficient for detection. First, foldback detection is often challenging with short-read sequencing. This is particularly true when the breaks occur in repetitive or low-complexity regions, or if there is a large distance between the ends of fusing sister chromatids. Second, many non-BFB amplifications, including ecDNA, present foldbacks and sharp ladder-like copy number changes. We resolved the first problem by using longer reads. Specifically, we used optical genome maps which provide megabase length scaffolds that detect foldbacks reliably, while linking multiple foldbacks in a single OGM molecule. To resolve the second issue, we developed an algorithm for measuring the likelihood of a BFB amplicon sharing the copy number segmentation and the foldback structures to refine the detection signature. Our method starts by extracting out the processed data into a copy number vector and foldback read vectors. Therefore, it can be extended to nanopore and other long read modalities, simply by changing the initial processing steps, and we plan to do that in future research.

Our model infers a mechanism of BFB formation and structural evolution that is based on the pattern of copy numbers and foldbacks inferred by analyzing an extensive amount of data from cancer cell lines and clinical samples. Future studies to prospectively model BFB formation and watch its evolution over time will be important to confirm the proposed mechanism and better understand its implications.

It is important to note that OM2BFB will not detect all BFBs. For example, a single cycle of bridge formation and breakage will lead to a very weak signal that will not be detectable. We would like to claim that the oncogenic effects of BFB are seen largely after a few cycles have sufficiently increased the copy number of the oncogene, and in these cases, the signal should be strong enough to be detectable by OM2BFB. However, additional studies will be needed to confirm these assertions. Similarly, telomere loss followed by fusion could lead to bridge formation may occur in non-homologous chromosomes[22], and additional structural variants may lead to translocation to non-native sites. Our current methods will need to be extended to handle these cases. When applied to short-read (wgs) data, our methods use similar abstractions, and show similarly high precision, but have reduced sensitivity due to weaker detection and phasing of SVs.

Nevertheless, the methods we developed here allowed us to systematically explore BFB amplifications in thousands of cancer samples and contrast them with other focal amplifications, including ecDNA. The identification of hundreds of BFB amplifications allowed us to identify cancer subtypes where BFB amplification was prevalent. BFB amplifications are found in many cancers where ecDNA have been observed. However, they are rare in cancers of the brain and central nervous system, where ecDNAs are very common. The reasons for this behavior are not entirely clear. An interesting related observation is that lower-grade gliomas tend to have longer telomeres and are less susceptible to telomere loss[49]. It is also plausible that fragile regions initiate BFB formation in specific regions, and not in others. However, our analysis of recurrent BFB amplicons did not support that hypothesis.

Somewhat surprisingly, we found that the genes amplified by BFB cycles are often also amplified on ecDNA and that cancer subtypes (and even single samples) with BFB amplification also show ecDNA amplification. However, some genes (e.g. *MYC, CDK4*) are predominantly amplified as ecDNA. More research is needed to clarify the mechanisms of ecDNA and BFB formation.

While ecDNA and BFB events amplify a similar subset of genes, they are structurally and functionally different. BFB amplicons have nearly twice the span of ecDNA amplicons. However, the localized BFB structure implies that it rarely co-amplifies distant oncogenes, unlike ecDNAs. The increased structural changes and the circularization in ecDNA change their conformation, enabling enhancer hijacking, and the alteration of their TAD boundaries. In contrast, BFBs, which do not have head to tail circularization, or extensive rearrangements, do not create as many novel interactions with distal regions. Increased oncogene expression in BFBs is more likely to be mediated by an increase in DNA copy number. Indeed the variability of gene expression after controlling for copy number is lower for BFB amplifications than on ecDNA. The gene expression programs are also markedly different, especially for immune response cells and for checkpoint genes controlling the immune response. These preliminary findings suggest that BFB(+) tumours may not be as immunosuppressive as ecDNA(+) tumours, and might be more amenable to checkpoint inhibition.

Anaphase bridge formation and bridge breakage have been shown to be a precursor to genome instability, including chromothripsis and ecDNA formation[23]. While true, our results also suggest that BFB cycles can lead to a stable amplification step, where only the native chromosome is impacted. We find that the pure BFB amplification studied here is often stable and shows much lower cell to cell

heterogeneity compared to ecDNA. This could make BFB amplicons more sensitive to targeted therapy and might anecdotally explain the success of anti-*Her2* therapy in *HER2*+ breast cancers, which are often driven by BFB amplifications[19].

However, BFB(+) cells can also acquire resistance by population shifts towards cells with alternative amplifications or increased amplification to compensate for the drug. Patient HN137 was initially responsive to anti-*EGFR* therapy but subsequently developed resistance[38]. While the primary cell line was sensitive to Gefitinib, the untreated metastatic cell line was resistant, and showed an increased copy number for *EGFR* in BFB amplification. We observed conserved BFB structures that amplified *EGFR* in both HN137Pri and HN137Met. Surprisingly, HN137Met showed two chromosomal foci, as well as a small number of ecDNA. These early data suggest some plasticity in BFB amplifications and raise the intriguing possibility that ecDNAs mediate the translocation of the BFB structure across chromosomes, or that increasing genomic instability resulted in ecDNA formation and translocation of the BFB.

HN137Met, but not HN137Pri, showed *YAP1* amplification on BFB. This likely explains the sensitivity of the Met line to YM155, an inhibitor of *BIRC5* and the Yap-Hippo pathway, and shows a shift in the clonality of cell populations between HN137Pri and HN137Met. In the larger cancer genome atlas data, we observed that patients with BFB(+) amplicons, but not ecDNA, had better survival outcomes in the initial period but the advantage was subsequently lost. While these results should be revisited and refined with larger data sets, they support the notion of BFB amplicons being more stable, and less adaptive relative to ecDNA, resulting in delayed onset of resistance.

Nearly 80 years after it was first discovered by Barbara McClintock in irradiated maize cells, the BFB cycle stands firm as an important and distinct mode of focal oncogene amplification in cancer. Identification and reclassification of cancers based on the mechanism of focal amplification will provide more insights into cancer pathology and treatment options.

## Methods
### Bionano optical genome mapping
Ultra-high molecular weight (UHMW) DNA was isolated from cells using a Bionano Prep SP Blood and Cell Culture DNA Isolation kit (#80042). In brief, about 1 million cells for each sample were lysed and digested in a mixed buffer containing Proteinase K, RNase A, and LBB lysis buffer following the manufacturer's instructions (Bionano Genomics). A Nanobind Disk was then added to the lysate to bind genomic DNA (gDNA) upon the addition of isopropanol. After washing, the gDNA was eluted and subjected to limited shearing to increase homogeneity by slowly pipetting up and down using standard 200 ul tips. The gDNA was then equilibrated overnight at room temperature to enhance homogeneity. 2 ul of gDNA aliquot was diluted in Qubit BR buffer and sonicated for 15 min before measuring concentrations with the Qubit dsDNA BR assay kit (Invitrogen Q3285). The UHMW gDNA was ready for labeling when the coefficient of variation of the Qubit reads were less than 0.3. 750 ng purified UHMW DNA was fluorescently labeled at the recognition site CTTAAG with the enzyme DLE-1 and subsequently counter-stained using a Bionano Prep DLS Labeling Kit (#80005) following manufacturer's instructions (Bionano Prep Direct Label and Stain (DLS) Protocol #30206). OGM was performed using a Saphyr platform. Calling of low allele frequency structural variants was performed using the rare variant analysis pipeline (Bionano Solve version 3.6) on molecules ≥ 150 kbp in length. The rare variant pipeline enables the detection of SVs occurring at low allelic fractions. Molecules were aligned to the GRCh38 reference, and clusters of molecules (≥3) indicating SVs were used for local assembly. Local consensus assemblies had high accuracy and were used to make final SV calls by realignment to the reference genome. Separately, based on coverage, the pipeline also generated a copy number profile that identified gains

and losses. Briefly, molecules were aligned to the GRCh38 reference to create a coverage profile that was then normalized based on OGM controls and scaled against a baseline defined at CN = 2 in autosomes (X and Y have a sex chromosome-specific baseline). Putative copy changes were segmented, and calls were generated. Entire chromosomal aneusomies were likewise defined in the CN algorithm. All genomics data was provided in an anonymized format with no personal identifiers.

## Formalization of BFB cycles

Denote a chromosomal arm using consecutive genomic segments A, B, C, D, starting from the centromere and going towards the telomere. The BFB cycle starts with chromosome breakage or a telomere loss, removing segment D. In a pure BFB cycle, where only a single chromosome is implicated, we could see a bridge formation, leading to the di-centric arm $ABC\underline{CBA}$, with the bar representing an inversion of the genomic segment. Subsequent breakage between B and A leads to a genome $ABC\underline{CB}$, which carries an inverted duplication, and a broken end, allowing for the process to repeat. A small number of BFB cycles lead to a highly rearranged genome. For example,

$$ABC \rightarrow ABC\underline{CB} \rightarrow ABC\underline{CBB} \rightarrow ABC\underline{CBB}\underline{BBCCB}$$

Sampling sequences from the BFB rearranged genome and mapping back to the human reference, we obtain a characteristic *copy-number vector* [1,6,4,0] denoting the copy numbers of segments A through D. Sampling reads from the junction of $\underline{BBCC}$, and mapping back to the reference results in *right-foldback* structures. Similarly, we obtain *left-foldback* structural variants, corresponding to reads sampled from $\underline{BB}$.

An abundance of foldback reads, together with a ladder-like copy number amplification is considered as a signature of BFB. We emphasize, however, that not every structure with an abundance of foldbacks and a ladder like amplification (e.g. $AB\underline{BBBBB}CCC$) can be explained using BFB cycles, and a more careful exploration is needed.

## The OM2BFB approach to detecting BFB

The main difference between existing methods and OM2BFB is that OM2BFB explicitly generates candidate BFB architecture that can be generated by a sequence of BFB cycles. In fact, OM2BFB generates multiple candidate BFB architecture. Finally, OM2BFB scores each candidate architecture using an algorithm to estimate the (negative log-) likelihood of BFB cycle formation. High likelihood (low scoring) reconstructions are output, along with the score as described below.

OM2BFB utilizes the following steps to detect BFB.
1. Use the Bionano pipeline to identify coverage, left, and right foldbacks.
2. Select candidate regions.
3. Enumerate candidate BFB architectures and compute the score.
4. Output the most likely (least scoring) candidate BFB architecture, and provide the BFB(+) or BFB(-) label based on the score cut-off

Step 2 adapts methods previously developed[32,33]. The other steps are described in detail below.

## SV calling including left and right foldbacks

Conceptually, define an optical genome map (OGM) as a sorted list of numeric values, representing the relative positions of labels on a fragment of DNA. These numeric lists can be generated for any collection of individual OGM molecules, assembled OGM molecules, or from in silico predicted label positions on the reference genome. We utilized OGM data from Bionano Genomics, inc. (bionano.com). All OGM samples were pre-processed using the Bionano Solve pipeline. The pipeline aligns and assembles OGM molecules into larger OGM

contigs, while also correcting the inter-label distances. We also used Bionano Solve to map the assembled reads to the hg38 genomic reference, call copy numbers and structural variants (SVs). The data from the Bionano Solve pipeline is abstracted through OGM label locations. Each label $i$ is covered by $V_i$ molecules. The inferred copy number for label $i$ is inferred and represented by $N_i$. Similarly, we denote set of molecules that supports left and right foldback SVs at label $i$ as $F_{Li}$ and $F_{r,i}$, respectively. Thus the data is presented as a tuple $(N, V, F_l, F_r)$ for each label i. The majority of OGM samples exhibited a coverage range of 100–300x for the diploid chromosome.

## Candidate region selection and parameterization

Starting with tuples output by the Bionano Solve pipeline, applies some initial filtering steps to identify candidate BFB amplicons. First, candidate labels with a copy number (CN) >3, indicative of amplification, and containing over 10% more foldbacks compared to the average are selected. Subsequently, it determines the span of the amplicon by linking pairs of consecutively selected labels based on two parameters: the distance (D) between the labels and the average copy number (E) of the two labels. Two consecutive regions are clustered if either $D < 1.5$ Mbp or $E > 7$. These thresholds were empirically determined through the analysis of real BFB datasets(Supplementary Data 1). Finally, single linkage clustering was used to identify candidate BFB regions.

## Enumerate candidate BFB architectures

Given an observed or estimated copy number vector ($C^o$), we adapt previously described method BFB-Finder[32,33] to solve the following:
1. If there exists a candidate BFB architecture $B$ whose induced copy number $C$ exactly matches ($C^o$), then return the copy number, left-foldback, and right-foldback vectors ($C, L, R$ respectively) induced by $B$.
2. Identify multiple candidate BFB architectures induced copy number $C$ apprximately matches ($C^o$) For each such candidate architecture, return the induced ($C, L, R$) vectors.

See details in an online appendix.

## OM2BFB scoring

Consider a triple ($C, L, R$), corresponding to a copy number vector, a 'left-foldback' vector, and a 'right-foldback' vector of a candidate BFB structure. OM2BFB computes a likelihood for ($C, L, R$) given input genomic data. Note that the input genomic data ($N, V, F_l, F_r$) is indexed over labels, while ($C, L, R$) are indexed over genomic segments (each containing a multitude of labels). Therefore, we introduce latent variables ($C^o, L^o, R^o$) referring to observed counts, to estimate a candidate structure

$$\max_{C,L,R} \sum_{C^o, L^o, R^o} Pr(N, V, F_l, F_r | C^o, L^o, R^o).Pr(C^o, L^o, R^o | C, L, R) \quad (1)$$

In order to estimate $Pr(C^o, L^o, R^o | C, L, R)$, we submit the observed vectors to BFB-Finder, which will generate the nearest BFB count vector and multiple candidate BFB sequences, each with their own foldback vectors. For $n$ segments, labeled ($i, ..., n$), the Discrepancy between the observed and estimated vectors is computed using

$$\Delta((C^o, L^o, R^o), (C, L, R)) = \sum_{i=1}^{n} \frac{|C_i^o - C_i|}{C_i} + \alpha * EuclideanDistance((L^o, R^o), (L, R)) + P_1 * F \quad (2)$$

Here, parameter $P_1$ denotes penalty for foldback discrepancies, and F equals the total number of foldback discrepancies described by

$$F = \{\#i \ s.t. R_i = 0, R_i^o \neq 0 \ or \ L_i = 0, L_i^o \neq 0\} \quad (3)$$

The likelihood of the of (C,L,R) is given by

$$Pr(C^o, L^o, R^o | C, L, R) \propto \exp(-\Delta((C^o, L^o, R^o), (C, L, R))) \quad (4)$$

Next, we estimated $Pr(N, V, F_l, F_r | C^o, L^o, R^o)$, to model the BionanoSolve out which is $(N, V, F_l, F_r)$ relative to region based copy numbers and foldbacks, given by $(C^o, L^o, R^o)$.

First, we segmented CN vector $N$ using the Circular Binary Segmentation (CBS)[50]. Note that density of labels is not uniform over the reference genome and segment length between two consecutive labels can vary. Therefore, we prepared a weight vector for the CBS algorithm. For each segment, between two consecutive labels, with length larger than 10Kbp we normalized the weight as 1 and for segments with length less than 10 Kbp, the weight was chosen as the segment length in bp divided by 10000. After segmenting with CBS algorithm, we smoothed the result as follows: for each segment $S_i$, if it is a short segment which means that if its length is less than 7 percent of total region length and triplet $(S_{i-1}, S_i, S_{i+1})$ was monotonically increasing or decreasing, segment $S_i$ was merged with the consecutive segment that was closest in copy number. The empirical smoothing removed very short segments associated with sharp increase or decrease in copy numbers. After merging these short segments, the CN vector $C^o$ was obtained.

Define $Z_1$ as the expected number of raw molecules that cover a label with copy number 1. Then, $Z_1 = \frac{\sum_i V_i}{\sum_i N_i}$, where the indexing is over all labels.

To estimate the CN of each foldback, we divided the number of supported molecules by $Z_1$. Hence,

$$R_S^0 = \frac{|\cup_{i \in s} F_{r,i}|}{Z_i}, L_S^0 = \frac{|\cup_{i \in s} F_{l,j}|}{Z_i} \quad (5)$$

The Discrepancy between the raw input and observed vectors was estimated using:

$$\Delta((N, V, F_l, F_r), (C^o, L^o, R^o)) = \sum_{s=1}^{n} \frac{\sum_{i \in s}^{m} |N_i - C_s^o|}{m} \quad (6)$$

where $m$ is number of labels covering by segment $s$. The discrepancy helps estimate the likelihood of $(C^o, L^o, R^o)$ using:

$$Pr(N, V, F_l, F_r | C^o, L^o, R^o) \propto \exp(-\Delta((N, V, F_l, F_r), (C^o, L^o, R^o))) \quad (7)$$

We used a Gibbs sampling approach where in each step, $(C_i^0, L_i^0, R_i^0)$ was sampled for segment $i$ using the observed label counts and modifying the observed label counts, merging two adjacent segments, or splitting two segments. At the end of the procedure, the OM2BFB method returns candidate BFB structures that best explain the observed OGM data, along with a score given by:

$$BFB - score = \frac{1}{n^c}[\Delta((C^o, L^o, R^o), (C, L, R)) + \Delta((N, V, F_l, F_r), (C^o, L^o, R^o))] \quad (8)$$

### Parameter selections

We optimized the parameters $P_1$, $\alpha$, and $c$ using a grid search to maximize the separation between known BFB (-) and BFB (+) simulated samples (Supplementary Fig. 25). We selected 90 BFB (-) and 90 BFB (+) cases for this optimization. As multiple combinations are likely to lead to optimal separation, we performed an initial search fixing $P1 = 1$, and testing $\alpha$ in the range (3,10) and $c$ in the range (0.8,1.2). The test revealed $\alpha = 7$. $c = 0.9$, as the optimum choices To confirm these values,

we fixed each pair of parameters, and varied the remaining parameter over a larger range.

First, we fixed $\alpha$ and $c$, and varied $P_1$ within the range of 0 to 10, calculating the F1 score for each value. The existence of $P_1$ significantly impacted the F1 score, although increasing $P_1$ did not consistently enhance it. Given the likelihood of missing a foldback in real datasets, we chose the minimal value, $P_1 = 1$ (Supplementary Fig. 25). A similar grid search was conducted for $\alpha$. With $c$ and $P_1$ fixed, we tested different values of $\alpha$. Consistent with our prior approach, $\alpha = 7$ yielded a near-maximum F1 score with a minimal value, leading to its selection. Lastly, for parameter $c$, we fixed $P_1$ and $\alpha$, finding that $c = 0.9$ provided almost the maximum F1 score (Supplementary Fig. 25). We kept these values for all future tests. As more experimental data becomes available, these parameters will be re-estimated.

### Visualization

The BFB reconstructions were visualized using a stylized format, as shown in Fig. 1B. The axes display genomic coordinates (x-axis) and copy-number (y-axis), with a separate track showing foldbacks. The proposed BFB structure can be read by marking the segments traversed by the blue line, starting from the centromeric end. Red transitions correspond to missing foldbacks required to explain the BFB structure. To avoid huge repetitions of a core structure, OM2BFB depicts only the core structure, along with the multiplicity of repetitions.

### Simulations

BFB molecules were simulated using a custom-developed tool named BFBSimulator, which accepted parameters defining the number of BFB cycles, a chromosome, and start and end positions. Additionally, optional parameters were implemented to allow control over the mean and distribution of BFB segments, along with parameters governing deletion lengths and distribution within the BFB structure and foldback SVs. The tool is publicly available here https://github.com/poloxu/CSE280A_BFBSimulator

Three distinct types of cases were simulated, each representing different complexity levels:

1. **Simple**: Characterized by low segment count, low copy number, no indels, and no deletion in folding regions.
2. **Intermediate**: Regular segment number and copy number were used, along with deletion in folding regions, but no indels were present.
3. **Complex**: This case included regular segment number and copy number, folding region deletions, and indels.

The output from BFBSimulator was formatted as a fasta file. Subsequently, OMSim was employed to simulate optical genome map (OGM) molecules from the fasta output. Parameters recommended by OMSim's developer were used for simulating the DLE-1 enzyme, but to bypass the assembly of OGM molecules, we simulated long OGM molecules (average length of 10 Mbp) and high accuracy and aligned them to the reference genome. The final step involved using the optical genome map alignment tool FaNDOM[31] to align the molecules to the reference genome and do the SV calling. In the analysis of OM2BFB, alignments, SV call, and CNV call were essential. Alignments and SV calls from simulated molecules were obtained from FaNDOM and used to create simulated CNV calls. Taking the ground truth CNV values for each genomic segment from the simulated BFB structure we made two different cases. One case segments with exact copy number and contains sharp alterations in segment copy number at segments border, but in the second case, the copy number segmentation boundaries are more gradual, as they appear in real data (Supplementary Fig. 26).

For simulating BFB negative cases, the focal amplification simulator, ecSimulator[8](https://github.com/AmpliconSuite/ecSimulator)

was employed with different sets of settings to cover simple and complex extrachromosomal DNA (ecDNA) (998 cases) and chromothripsis (200 cases) structures in terms of SV rates, with or without duplication inversions (Supplementary Fig. 27). Upon obtaining the fasta file from ecSimulator, the rest of the pipeline, including OGM molecule simulation, contig alignment, and CNV call generation, remained consistent with the process used for BFB positive cases.

**Amplicon classification and BFB detection.** We utilized AmpliconClassifier (AC) (version 0.4.11, available from AmpliconSuite at https://github.com/AmpliconSuite/AmpliconClassifier) to identify BFB cycles. AC takes as input the AA breakpoint graph file encoding genomic segment copy numbers and SV breakpoint junctions, as well as the AA cycles file encoding decompositions of the AA graph file into overlapping cyclic and/or non-cyclic paths. Each path and cycle is weighted by the genomic CN it represents. AmpliconClassifier uses multiple heuristics to call BFBs, as described earlier[39]. The salient parts are described below. First AC filters short paths (length < 10 kbp), paths which significantly overlap low-complexity or repetitive regions, and paths which overlap regions of the genome of low copy number.

AC first assesses non-filtered paths for the presence of BFB cycles using heuristics determined from manual examination of BFB-like focal amplifications in the FHCC cohort and focal amplifications in previous studies[5,28]. AC computes a few relevant statistics: (a) the fraction '$f$' of breakpoint graph discordant edges which are foldback, and the paired-ends have a genomic distance <25 kbp. AC next identifies decomposed paths containing foldback junctions between segments, and using all paths computes the set of consecutive segment pairs in the paths where the two boundaries of the segments together form a foldback junction. Each segment pair is assigned its own weight equal to the decomposed copy count of the path. If the proportion of BFB-like segment pairs over all segment pairs in all paths is less than 0.295, then the amplicon is not considered to contain a BFB. Furthermore, if the total weights of pairs which are "distal" (not foldback and > 5kbp jump between endpoints) divided by the total weight of all pairs is greater than 0.5, the amplicon is not considered to contain BFB. Lastly, if the total decomposed CN of all pairs is <1.5, or if the total number of foldback segment pairs is <3, or $f$ < 0.25, or the decomposed CN weight of all BFB-like paths divided by the CN weight of all paths <0.6, or the maximum genomic copy number of any region in the candidate BFB region is <4, the amplicon is not considered to contain a BFB. If the amplicon has not failed any of these criteria, a BFB(+) status is assigned, and the BFB-like cycles (decomposed paths with a BFB foldback) are separated before additional fCNA detection inside the amplicon region.

## Interphase FISH analysis

In Interphase FISH analysis, the input is an image containing multiple interphase nuclei stained with DAPI and with fluorescent painting of a probed target. The output is a binary segmentation image per probe, with the regions in the image predicted as amplifications are set with value 1 and the background is set with value 0. Our tool returns a binary segmentation image per probe, and each channel (other than DAPI) in the output image is analyzed independently of the other gene probes and other images. The high level steps are as follows:

1. Nuclear segmentation to identify the pixels corresponding to each intact nucleus.
2. Identification and quantification of FISH foci for each nucleus.

   These steps are described below.

## FISH nuclear segmentation

The chromosomes are unraveled during interphase and occupy much of the nuclear volume. Therefore, the DAPI stain helps separate nuclear regions from the cytoplasmic ones. In patient tissue, however, the

nuclei are tightly packed and difficult to resolve into individual nuclei. We applied NuSeT to perform nuclei segmentation[51]. We used a min_score of 0.95, an nms threshold of 0.01, and a scale ratio of 0.3 for all image datasets. For the cell lines with a mix of interphase and metaphase cells (COLO320DM_MYC and HCC827_EGFR), we used a nucleus size threshold of 5000 to prevent individual chromosomes from being classified as nuclei. For the interphase only cell lines (354PDX_ERBB2, 355PDX_ERBB2, 727PDX_ERBB2, Ni8PDX_ERBB2, Ni17PDX_ERBB2, MSTO211H_MYC, COLO320HSR_MYC), we used a nucleus size threshold of 500. Additionally, we applied the min-cut algorithm to convert NuSeT's binary segmentation output to an instance segmentation. In order to separate neighboring nuclei, for each connected component in the image, we created a pixel graph with 4-connectivity only for pixels with the nuclei binary segmentation value. We looked for pixel centers by convolving the image with a uniform filter, and looked for the minimum number of edges to remove to separate 2 centers in the same connected component in the binary mask. We examined all nuclei with an area larger than 1.25 times the median nuclei size, and separated nuclei with a flow limit of 60.

## Number of FISH foci

To quantify the number of FISH foci, we convolved the original image with a normalized gaussian kernel to determine which pixels have high local intensity. We normalized the gaussian kernel sigma by the median nuclei size for each image. For the median nuclei size of 2500, we used a sampled gaussian kernel with a standard deviation of 3 pixels, and a size of 7 by 7 pixels. After convolving, we applied a threshold of 15 / 255 pixel brightness. Then, to filter out low brightness noise, we have a brightness threshold of 100 / 255 on the original image, and set the minimum spot size to 7 pixels. To exclude multiple merged nuclei from impacting the FISH foci counts, we removed all nuclei in the top 10 percentile for nuclei area for each sample. Additionally, for cell lines with greater than 50 remaining nuclei, we randomly selected the FISH spot counts from 50 nuclei without replacement in the violin plot.

## Test for fragile regions

we investigated the potential influence of positive selection in Breakage-Fusion-Bridge (BFB) regions through a permutation-like test (one-tailed). Seven genomic regions, each containing a minimum of 7 BFBs, were selected. For each region, we divided it into 10 non-overlapping 1 Mb windows positioned telomeric to an oncogene. Letting $t_i$ represent the number of BFBs in window $i$, $s$ denoting the total number of BFBs in the region, and $t_m$ representing the maximum number of BFBs observed in a single window. The p-value was calculated as the probability of randomly distributing $s$ BFBs into the 10 windows such that some window contained at least $t_m$ BFBs. This probability is equal to

$$P = \frac{-\sum_{m=1}^{m=\lfloor s/t_m \rfloor} (-1)^m \binom{10}{m} \binom{s-mt_m+10-1}{10-1}}{\binom{s+10-1}{10-1}} \quad (9)$$

## H3K27ac HiChIP data

We downloaded the hg19-aligned and processed H3K27ac HiChIP for GM12878 from Mumbach et al.[52] and COLO320DM from Hung et al.[42]. We reused the WGS data and cycle structure for COLO320DM using previously reported work[12,42], where the structure of the *MYC*-amplified ecDNA was fully resolved. We aligned the COLO320DM WGS to hg19 and ran AA (version 1.3.r5) followed by AC (version 0.5.4) to get the list of SVs involving the *MYC*-amplified ecDNA and the BFB-driven focal amplification.

## Distal chromatin interaction identification

We used NeoLoopFinder[46] version 0.4.3 to search for chromatin interactions on the *MYC*-amplified ecDNA and the BFB-driven focal

amplification on chr1 on CN-corrected HiChIP matrices at resolutions 5 k and 10 k. NeoLoopFinder, by default, computes a genome-wide CN profile and a collection of CN segments from an input contact matrix, and then balances the matrix with a modified ICE procedure by taking the CN segments as input. Given a list of candidate SVs (potentially from other sources, e.g., WGS or OM), it then reconstructs local assemblies representing a chain of one or more SVs from the input list, by shifting or flipping the submatrices according to the coordinates and orientations of the SVs. Therefore, we supplied NeoLoopFinder with a collection of SV breakpoints derived from AA. Additionally, we augmented the assemblies constructed by NeoLooFinder with the collection of local assemblies from the known ecDNA structure in Hung et al. as follows. Because NeoLoopFinder does not accept assemblies with duplicated segments, we broke the 4.3 Mbp ecDNA cycle into all possible longest paths of at least 2 non-overlapping segments. This resulted in 6 distinct assemblies, which were provided as input to NeoLoopFinder to search for chromatin interactions (Supplementary Data 7) in addition to the local assemblies constructed above. NeoLoopFinder identified interactions from these HiChIP matrices at different resolutions, and merged the results. It outputs two types of interactions: 'loops,' which represent interactions on a single genomic segment, and 'neo-loops,' representing interactions on two different genomic segments, brought together by an SV.

As NeoLoopFinder does not support SVs which induce overlapping segments, including foldback SVs, no neo-loops spanning a foldback SV were reported. However, we identified loops by providing NeoLoopFinder with a BFB-driven local assembly on chr1 that fully covered the foldback SVs.

On the control cell line GM12878, whole genome distal chromatin interactions were identified using the method in Salameh et al.[53] (which was called internally by NeoLoopFinder) and with the same set of resolutions (5k, 10k).

We compared the number of interactions identified by NeoLoopFinder at different distances, normalized by the size of the focal amplifications, for the *MYC*-amplified ecDNA, the BFB-driven amplification and the control cell line GM12878. For GM12878 we included all chromatin interactions and normalized the count at a particular distance by the total genome length. The distribution can be visualized in Fig. 4C. In order to assess the statistical significance of the differences between the distributions of BFB and ecDNA, we employed the Peacock test[54], a multidimensional version of the Kolmogorov-Smirnov test. This test is specifically designed to analyze random samples defined in two or three dimensions.

**Focal amplification classification from paired-end WGS.** We utilized previously published AmpliconArchitect outputs[5] from TCGA tumour samples and deployed AmpliconClassifier (AC) version 0.4.11 (default settings) to predict presence of BFB in the samples. AC also annotated structures based on gene contents, copy number, and reported an entropy-based complexity score for the focal amplification, using methods described in Luebeck et al.[39].

## Cell culture

Cell lines were purchased from ATCC or Sekisui Xenotech. BT474 was maintained in ATCC Hybri-Care Medium (ATCC, #46-X) with 10% FBS and 1% PSQ. COLO320DM, COLO320HSR and PC3 were cultured in DMEM (Corning, #10-013-CV) with 10% FBS and 1% PSQ. HARA, H2170, H460, HCC827, OVCAR3 and SJSA1 were cultured in RPMI-1640 (ATCC modification) (Gibco, #A1049101) with 10% FBS and 1% PSQ. THP1 was cultured in ATCC-formulated RPMI-1640, supplemented with 0.05 mM 2-mercaptoethanol and 10% FBS. SNU16M1 was cultured in F12/DMEM supplemented with 10% FBS and 1% PSQ. HCC827 naive, drug resistant (HCC827 ER, HCC827 LR) and drug removed lines (HCC827 ERDR) were cultured in RPMI-1640 (ATCC modification) (Gibco, A1049101) with 10% FBS and 1% PSQ. 3 μM erlotinib and 1 μM lapatinib were added

to HCC827 ER and HCC827 LR respectively to maintain drug resistance. All cell lines were maintained in a 37 °C tissue culture incubator supplemented with 5% $CO_2$.

## Metaphase FISH

Metaphase FISH was collected for cultured cell lines. Cells were arrested in mitosis by KaryoMAX Colcemid (Gibco) treatment at 100 ng ml−1 for 4 h. Cells were washed once in 1X PBS and incubated in 0.075 M KCl hypotonic buffer for 20 min at 37 °C. Carnoy fixative (3:1 methanol:glacial acetic acid) were added to fix and wash cells for a total of 3 times. Cells were dropped onto a humidified glass slide and completely air dried. The slides were equilibrated briefly in 2X SSC buffer, followed by ethanol dehydration in ascending ethanol concentrations (70%, 85% then 100%) for 2 min each. FISH probes diluted in hybridization buffer (Empire Genomics) at 1:6 ratio were added to the sample and sealed with a coverslip. DNA strands were denatured at 75 °C for 3 min, followed by hybridization at 37 °C overnight in a dark humidified chamber. Coverslips were removed and samples were washed in 0.4X SSC for 2 min, followed by 2X SSC 0.1% Tween 20 for 2 min, and rinsed briefly in 2X SSC. DAPI (50 ng/mL) was used to stain nuclei for 2 min, and washed briefly in ddH$_2$O. Air-dried samples were then mounted with ProLong Diamond Antifade (Invitrogen, #P36931) and cured for at least 4 h prior to imaging. Images were acquired on a Leica DMi8 widefield microscope on a 63X oil lens.

FISH probes were obtained from Empire Genomics (*CCND1*, #CCND1-20-GR; CCNE1, #CCNE1-20-RE; FANCG, #FANCG-20-GR; FGFR1, #FGFR1-20-RE; FGFR2, #FGFR2-20-RE; *MYC*, #MYC-20-GR; Myc-L1, #MYCL1-20-RE; PAK1, #PAK1-20-RE; *PDHX*, #PDHX-20-RE; Chromosome 9 Control Probe, #CHR9-10-RE; Chromosome 11 Control Probe, #CHR11-10-AQ), OGT CytoCell (*EGFR* Amplification Probe, #LPS 003; MDM2 Amplification Probe, #LPS016), Metasystem (XCEP1, #D-0801-050-FI; XCEP10, #D-0810-050-FI) and Agilent (SureFISH chromosome 19 probe, #G101075-85501).

## Tissue FISH

Tissue FISH were acquired for BCBM samples and cancer derived cell lines HN137Pri and HN137Met. All studies were approved by Institutional Review Board (IRB) approved protocols (DFCI IRB 93-085 and 10−417) and the Singhealth Centralized Institutional Review Board (CIRB 2007/441/B). FFPE slides were deparaffinized by two exchanges of xylene for 5 min each. The slides were rehydrated in 100% ethanol for 5 min, followed by 70% ethanol wash for another 5 min. Slides were briefly rinsed in ddH$_2$O and immersed in 0.2 N HCl for 20 min. Antigen retrieval was performed by immersing the slides in 10 mM citric acid solution (pH 6.0) and microwaved to reach a temperature at about 90−95 °C for 15 min. Slides were rinsed briefly in 2X SSC and digested with 1 μL Proteinase K (NEB, #P8107S) diluted in 100 μL TE buffer at room temperature for 1 min, and washed briefly in ddH$_2$O. Slides were then dehydrated in a series of ascending ethanol (70%, 85% and 100%) for 2 min each. FISH probes diluted in hybridization buffer (Empire Genomics) in 1:6 ratio were applied to the slide. Samples were denatured at 75 °C for 3 min and hybridized at 37 °C overnight in a dark humidified chamber. Slides were washed in 0.4X SSC (0.3% IGEPAL) at 40 °C for 5 min twice, followed by another wash in 2X SSC (0.1% IGEPAL) at room temperature for 5 min. Autofluorescence was quenched following the manufacturer's instructions of the Vector TrueVIEW Autofluorescence Quenching Kit (Vector Laboratories, #SP-8400-15). The slides were stained in DAPI (50 ng/mL) for 10 min, rinsed twice in 2X SSC and once in ddH$_2$O. Air-dried slides were mounted with Prolong Diamond Antifade (Invitrogen, #P36931) and cured overnight at room temperature. Images were acquired on a Leica DMi8 widefield microscope using a 63X oil objective, z-stack images were post-processed using Small Volume Computation Clearing on the LAS X thunder imager prior to generating max projections.

FISH probes were obtained from Empire Genomics (ERBB2, #ERBB2-20-GR; KCTD8, #KCTD8-20-GR; PAX6, #PAX6-20-RE; RP11-1029O18 FISH Probe-Green; Chromosome 4 Control Probe, #CHR4-10-RE; Chromosome 11 Control Probe, #CHR11-10-AQ) and KromaTiD (subCEP CHR 5p Pinpoint FISH Probe, #CEP-0009-C; subCEP CHR 17p Pinpoint FISH Probe, #CEP-0033-D).

## WGS sample and library preparation
WGS was acquired for cell line HCC827_ER. gDNA was extracted with the Qiagen DNA Mini Kit following manufacturer's instructions. 250 ng of gDNA were used to generate sequencing libraries using the NEBNext Ultra II FS DNA Library Prep Kit, following the manufacturer's protocol to yield the final library with a size distribution between 320-470 bp.

## Patient-derived xenografts for Breast Cancer with Brain Metastases (BCBM) samples
Acquisition of human samples was approved by the Institutional Review Board (IRB) protocols (DFCI IRB 93-085, 10-417, 18-296). Informed consent was obtained from breast cancer patients. Fresh brain metastases were obtained from female breast cancer patients at Brigham and Women's hospital under IRB-approved protocol (DFCI IRB 93-085 and 10-417). PDXs were established via intracranial injection as described previously[36,37]. Tumor samples were dissociated using Collagenase/Hyaluronidase and Accutase, and cells were injected into the striatum of 6–10 weeks old female ICR-SCID female mice (Taconic, IcrTac:IR-Prkdcscid). Female mice were chosen to match the patient-derived nature of the samples. Mice were then monitored daily for health conditions. For intracranial tumors, the maximal tumour size/burden was not exceeded. The maximal tumour size/burden was not defined by size but by endpoints related to health conditions, including neurological symptoms. All procedures were approved by Dana-Farber Cancer Institute Animal Care and Use Committee in compliance with NIH animal guidelines.

## Model cell lines for head and neck cancer samples
Tumour samples were obtained from patients post surgery after obtaining informed patient consent in accordance with SingHealth Centralized Institutional Review Board (CIRB 2007/441/B and CIRB: 2014/2093/B). Model cell lines were generated as described previously[38,55].

## Bionano optical mapping (OGM) sample preparation
OGM data was acquired for Breast Cancer with Brain metastases (BCBM) samples. All procedures were approved by Dana-Farber Cancer Institute in compliance with NIH animal guidelines. OGM data was acquired for Head and Neck Cancer samples HN137Pri and HN137Met. The procedures were approved by the Singhealth Centralized Institutional Review Board (CIRB 2007/441/B). Cell cultures were maintained as described above and a total of 1.5 M cells were pelleted in a 1.5 mL microfuge tube. The cells were washed twice in 1X PBS and snap-frozen at -80 °C prior to shipment to Bionano Genomics for 100x Human Genome Sample Analysis.

## Reporting summary
Further information on research design is available in the Nature Portfolio Reporting Summary linked to this article.

## Data availability
The genomic data utilized in this study is sourced from various repositories and studies, adhering to the principles of open science and data sharing. The OGM data from BCBM samples (Methods) is available at PRJNA1022500. All procedures were approved by Dana-Farber Cancer Institute Animal Care and Use Committee. OGM data from Head and Neck Cancer cell-lines HN137Pri and HN137Met is available at PRJNA1022500. All procedures were approved by Singhealth

Centralized Institutional Review Board (CIRB 2007/441/B). OGM data for Medulloblastoma cell-lines was downloaded from PRJNA1011359 as reported in this study[56] [https://doi.org/10.1038/s41588-023-01551-3]. OGM data for cancer cell-lines was acquired and is accessible at PRJNA1022500. WGS data for HCC827, HCC827LR, and HCC827DR are available under accession number PRJNA338012 from a previous publication[4] [https://doi.org/10.1038/nature21356]. The HCC827-ER WGS is available from SRR31728042 [https://www.ncbi.nlm.nih.gov/sra/SRX27090373]. Amplicon Architect output for HCC827 supporting Fig. 4 are publicly available at https://ampliconrepository.org/project/673e3bd2642565afc9a37c56. AmpliconArchitect outputs for TCGA were obtained from a previous study[5] [https://doi.org/10.1038/s41588-020-0678-2] and they are publicly available at https://ampliconrepository.org/project/655bda68bba7c92509522479. AmpliconClassifier (AC) calls on CCLE data were downloaded from https://ampliconrepository.org/project/6580f373ea940f33361428ba. AmpliconArchitect outputs for the Barrett's esophagus/esophageal cancer dataset were obtained from previously published study[39] [https://doi.org/10.6084/m9.figshare.21893826.v1]. Source data are provided with this paper.

## Code availability
The following tools are available online: OM2BFB[57]: https://github.com/siavashre/OM2BFB under BSD 2-Clause License. BFB-Finder: https://github.com/shay-zakov/BFBFinder.

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

## Acknowledgements

This work was delivered as part of the eDyNAmiC team supported by the Cancer Grand Challenges partnership funded by Cancer Research UK (CGCATF-2021/100012 [P.S.M., H.Y.C.]; CGCATF-2021/100025 [V.B.])

and the National Cancer Institute (OT2CA278688 [P.S.M., H.Y.C.]; OT2CA278635 [V.B.]);U24CA264379, R01GM114362 (V.B.). This work was supported in part by BCRF-24–179 (to J.J.Z.),NIH R35 CA210057 (to J.J.Z.), and P50 CA168504. T.G.P NIH grants P01 CA91955 and P30 CA015704. H.Y.C. is an Investigator of the Howard Hughes Medical Institute. GX was supported by a postdoctoral fellowship from the Paul F. Glenn Center for Biology of Aging Research at the Salk Institute. The authors would like to acknowledge Prof Rebecca Fitzgerald (University of Cambridge) and the infrastructure supported by Cancer Research UK (CRUK) on behalf of the OCCAMS consortium for providing data and feedback used in this study.

## Author contributions

S.R. designed and implemented all software and analyses for this study and co-wrote the manuscript. I.W. generated all FISH data and contributed to the interpretation of results. A.G. generated additional FISH data. J.N. prepared breast cancer samples with breakpoint enrichment. J.L. adapted the Amplicon Classifier code for this study's purposes. K.Z. and B.C. performed experiments contrasting chromosomal composition in resistant cell lines. G.P. and U.R. developed software for analyzing FISH data. L.K. created simulation tools for BFB examination. G.X., C.C., J.A.L., and F.Y. and Q.J. acquired OGM sequences from selected samples. D.M., S.X.T., C.E.L.C., and R.D. collected matched OGM data from paired primary and metastatic samples. K.H. performed FISH experiments to identify cell line breakpoint patterns. A.C. developed software for selecting cell lines enriched with extrachromosomal DNA. A.W.C.P. provided expertise in Bionano SV calling and data analysis. L.A. and T.G.P. provided access to Barrett's Esophagus patient samples. F.B.F. supplied drug-resistant cell lines and associated data. J.M. provided access to cell line data with double stranded breaks and breakage fusion bridge formation. H.Y.C. contributed HiChIP data and provided feedback on results interpretation. J.Z. provided breast cancer brain metastases (BCBM) samples and assisted in manuscript preparation. J.A.L. and R.D. carefully read the manuscript, providing valuable feedback. P.S.M. and V.B. co-designed the scope of the study and co-wrote the manuscript.

## Competing interests

V.B. is a co-founder, consultant, SAB member and has equity interest in Boundless Bio and Abterra, and the terms of this arrangement have been reviewed and approved by the University of California, San Diego in accordance with its conflict of interest policies. P.S.M. is a co-founder, chairs the scientific advisory board (SAB) of and has equity interest in Boundless Bio. P.S.M. is also an advisor with equity for Asteroid Therapeutics and is an advisor to Sage Therapeutics. H.Y.C. is a co-founder of Accent Therapeutics, Boundless Bio, Cartography Bio, and Orbital Therapeutics, and is an advisor to 10X Genomics, Arsenal Biosciences, Chroma Medicine, and Spring Discovery. J.J.Z. is co-founder and direc-

tor of Crimson Biopharm Inc. and Geode Therapeutics Inc. LBA is a co-founder, CSO, scientific advisory member, and consultant for io9, has equity and receives income. The terms of this arrangement have been reviewed and approved by the University of California, San Diego in accordance with its conflict of interest policies. LBA is a compensated member of the scientific advisory board of Inocras. LBA's spouse is an employee of Hologic, Inc. LBA declares U.S. provisional applications with serial numbers: 63/289,601; 63/269,033; 63/366,392; 63/412,835 as well as international patent application PCT/US2023/010679. LBA is also an inventor of a US Patent 10,776,718 for source identification by non-negative matrix factorization. K.H. is a former employee of Boundless Bio, Inc. The remaining authors declare no competing interests.

## Additional information

**Siavash Raeisi Dehkordi**[1,25], **Ivy Tsz-Lo Wong**[2,3,25], **Jing Ni**[4,5,25], **Jens Luebeck**[1], **Kaiyuan Zhu**[1], **Gino Prasad**[1], **Lena Krockenberger**[1], **Guanghui Xu**[6], **Biswanath Chowdhury**[1], **Utkrisht Rajkumar**[1], **Ann Caplin**[1], **Daniel Muliaditan**[7], **Aditi Gnanasekar**[2,3], **Ceyda Coruh**[6,8], **Qiushi Jin**[9], **Kristen Turner**[10], **Shu Xian Teo**[11], **Andy Wing Chun Pang**[12], **Ludmil B. Alexandrov**[13,14,15], **Christelle En Lin Chua**[11], **Frank B. Furnari**[16], **John Maciejowski**[17], **Thomas G. Paulson**[18], **Julie A. Law**[6,19], **Howard Y. Chang**[20,21], **Feng Yue**[9,22], **Ramanuj DasGupta**[7,23], **Jean Zhao**[4,5] ✉, **Paul S. Mischel**[2,3] ✉ & **Vineet Bafna**[1,24] ✉

[1]Department of Computer Science and Engineering, University of California San Diego, San Diego, CA, USA. [2]Department of Pathology, Stanford University School of Medicine, Stanford, CA, USA. [3]Sarafan ChEM-H, Stanford University, Stanford, CA, USA. [4]Department of Cancer Biology, Dana-Farber Cancer Institute, Boston, MA 02215, USA. [5]Department of Biological Chemistry and Molecular Pharmacology, Harvard Medical School, Boston, MA 02115, USA. [6]Plant Molecular and Cellular Biology Laboratory, Salk Institute for Biological Studies, La Jolla, CA 92037, USA. [7]Genome Institute of Singapore (GIS), Agency for Science, Technology and Research (A*STAR), 60 Biopolis Street, Genome, Singapore 138672, Republic of Singapore. [8]ClearNote Health, San Diego, CA 92121,

USA. [9]Department of Biochemistry and Molecular Genetics, Feinberg School of Medicine Northwestern University, Chicago, IL, USA. [10]Boundless Bio, San Diego, CA, USA. [11]Singapore Nuclear Research and Safety Initiative, National University of Singapore, Singapore 138672, Republic of Singapore. [12]Bionano Genomics, San Diego CA92121, USA. [13]Moores Cancer Center, UC San Diego Health, La Jolla, CA, USA. [14]Department of Cellular and Molecular Medicine, University of California at San Diego, La Jolla, CA, USA. [15]Department of Bioengineering, University of California at San Diego, La Jolla, CA, USA. [16]Department of Medicine, University of California at San Diego, La Jolla, CA, USA. [17]Molecular Biology Program, Memorial Sloan Kettering Cancer Center, New York, NY, USA. [18]Translational Science and Therapeutics Division, Fred Hutchinson Cancer Center, Seattle, WA, USA. [19]Division of Biological Sciences, University of California, San Diego, La Jolla, CA 92093, USA. [20]Center for Personal Dynamic Regulomes, Stanford University, Stanford, CA, USA. [21]Howard Hughes Medical Institute, Stanford University School of Medicine, Stanford, CA, USA. [22]Robert H. Lurie Comprehensive Cancer Center of Northwestern University, Chicago, IL, USA. [23]School of Cancer Sciences, University of Glasgow; Senior Group Leader, CRUK Scotland Institute, Garscube Estate, Switchback Road, Glasgow G61 1BD, UK. [24]Halıcıoğlu Data Science Institute, University of California at San Diego, La Jolla, CA, USA. [25]These authors contributed equally: Siavash Raeisi Dehkordi, Ivy Tsz-Lo Wong, Jing Ni. ✉e-mail: Jean_Zhao@dfci.harvard.edu; pmischel@stanford.edu; vbafna@ucsd.edu

