## [Peer Review File · Nature Communications]

Breakage fusion bridge cycles drive high oncogene number with moderate intratumoural heterogeneity

Corresponding Author: Dr Vineet Bafna

Version 0:

Reviewer comments:

Reviewer #1

(Remarks to the Author)

I thank the authors for their thoughtful and detailed rebuttal to my comments. I am overall impressed with their rebuttal and happy with the edits to the manuscript, and I am happy to say that these now address all my major concerns.

Just one small comment:

In first line of the results – they state that “BFBs start with a double strand break and telomere loss/erosion”, heavily paraphrasing the nice explanatory paragraphs in the cited reference (which is itself a review which presents the current putative priming event, rather than a research article demonstrating this fact). If space allows, I would consider slightly expanding and softening this definitive statement, and conveying that this remains a working hypothesis.

E.g. “A likely BFB-originating mechanism is the fusion of sister chromatids after a double strand break. This can generate dicentric chromosomes that are subject to iterative cycles of breakage and fusion (ref)”. Or similar.

(Remarks on code availability)

Reviewer #2

(Remarks to the Author)

The authors have addressed my concerns and I have nothing to add.

(Remarks on code availability)

REVIEWERS' COMMENTS

Reviewer #1 (Remarks to the Author):

I thank the authors for their thoughtful and detailed rebuttal to my comments. I am overall impressed with their rebuttal and happy with the edits to the manuscript, and I am happy to say that these now address all my major concerns.

Just one small comment:

In first line of the results – they state that “BFBs start with a double strand break and telomere loss/erosion”, heavily paraphrasing the nice explanatory paragraphs in the cited reference (which is itself a review which presents the current putative priming event, rather than a research article demonstrating this fact). If space allows, I would consider slightly expanding and softening this definitive statement, and conveying that this remains a working hypothesis.

E.g. “A likely BFB-originating mechanism is the fusion of sister chromatids after a double strand break. This can generate dicentric chromosomes that are subject to iterative cycles of breakage and fusion (ref)”. Or similar.

Response: We thanked the reviewer and changed the first line of result section to this:

A likely BFB-originating mechanism is the fusion of sister chromatids after a double strand break¹⁸. This can generate dicentric chromosomes that are subject to iterative cycles of breakage and fusion terminating with telomere restoration^{6,19,30}.

Reviewer #2 (Remarks to the Author):

The authors have addressed my concerns and I have nothing to add.